



# How montane forests shape snow cover dynamics across the central European Alps

Vincent Haagmans[1,2], Giulia Mazzotti[1,3,4], Clare Webster[1,5,6], Tobias Jonas[1]

[1]WSL-Institute for Snow and Avalanche Research SLF, Davos, Switzerland
[2]Department of Civil, Environmental and Geomatic Engineering, ETH Zürich, Zürich, Switzerland
[3]Université Grenoble Alpes, INRAE, CNRS, IRD, Grenoble INP, IGE, Grenoble, France
[4]Université Grenoble Alpes, Université de Toulouse, Météo-France, CNRS, CNRM, Centre d'Études de la Neige, St. Martin d'Hères, France
[5]Department of Geosciences, University of Oslo, Oslo, Norway
[6]Department of Geography, University of Zurich, Zürich, Switzerland

*Correspondence to*: Vincent Haagmans (vincent.haagmans@slf.ch)

**Abstract.** A substantial fraction of seasonal snow is stored in mid-latitude montane forests, serving as an essential temporal water storage. Across vast areas, snow cover dynamics are the result of processes equally controlled by forest structure, topography, climate, and weather variability. As data availability has limited our ability to
disentangle how these four key controls interact across landscapes within complex topography, most forest snow studies have focused on only one or two of the controls. In this study, we employed the process-based FSM2oshd forest snow model framework for an in-depth analysis of the current state of forest snow water resources across the central European Alps. Over the 8 years analysed, forest snow accounted for 20–30 % of the total snow storage in midwinter. In the various effects of existing forest cover on snow, pronounced differences were found depending
on elevation, aspect, region, and year. While the presence of forest usually led to a decrease in peak SWE, it decelerated snowmelt, often leading to a later snow disappearance date, particularly on south-facing slopes. However, variability between years and regions was strong enough to shift or even reverse such trends, where snow-scarce years accentuated relative differences in the effects of forests on snow cover. With forest disturbances projected to increase and snow storage to further decline, enhanced complexity of snow cover dynamics in montane
forests is to be expected. This places more emphasis on understanding how the effects of key controls such as forest structure, topography, and weather interact.

## 1    Introduction

Forest and seasonal snow co-occur on about 20 % of the surface area of mid-latitude mountain ranges (Deschamps-Berger et al., 2025), which serve as essential temporal water storage for downstream populated regions (Immerzeel
et al., 2020; Viviroli et al., 2007). In the European Alps, for example, snowmelt is the primary seasonal runoff signal (Beniston, 2012) and accounts for up to 40 % of annual runoff in Switzerland (FOEN, 2021)**.** Therefore, understanding how forest impacts snow cover across mountain ranges is important for water resource and forest resource management (Dickerson-Lange et al., 2021, 2023), streamflow forecasting (Pomeroy et al., 2012; Sun et al., 2018; Troendle, 1983), as well as ecological processes (Cooper et al., 2020; Sanmiguel-Vallelado et al., 2021).

During the snow accumulation period, interception of snow by forest canopies (Hedstrom and Pomeroy, 1998; Roth and Nolin, 2019) decreases below-canopy snow depth (Dharmadasa et al., 2023; Mazzotti et al., 2019a; Moeser et al., 2015) and snow water equivalent (Harestad and Bunnell, 1981; López-Moreno and Latron, 2008; López-Moreno and Stähli, 2008; Stähli and Gustafsson, 2006). As winter progresses*,* snow ablation is influenced by the presence of forest canopies that mask the forest floor from shortwave radiation and enhance longwave
radiation (Essery et al., 2008; Link et al., 2004; Lundquist et al., 2013; Mazzotti et al., 2019b; Sicart et al., 2004). This usually results in less radiative energy reaching the forest floor compared to open areas (Link et al., 2004; Strasser et al., 2011). Reduced wind speeds in forests also limit the magnitude of turbulent fluxes (Conway et al., 2018; Price, 1988; Roth and Nolin, 2017). These processes occur wherever forests and snow coexist, but their relative magnitude and net effect on forest snow resources vary depending on forest structure, topography, weather,
and climate (Lundquist et al., 2013; Roth and Nolin, 2017; Safa et al., 2021).

Forest structure exerts a first-order control on the strength of most forest-snow processes at the local site scale, while climate exerts a first-order control between regions (e.g., Broxton et al., 2020). Forest structure is highly heterogeneous, and the spatial scales over which different forest-snow processes act strongly vary. This results in spatiotemporal patterns of forest-snow processes that are highly complex at scales from the individual tree crown
(meter; Mazzotti et al., 2019a) to the site/plot scale (meter-hectare; Koutantou et al., 2022) and beyond (meter-



valley; Mazzotti et al., 2023). Furthermore, local climatic conditions influence the likelihood of different processes dominating the spatial variability in snow cover within a specific region. For example, denser canopies will intercept more snow than sparse canopies (Moeser et al., 2015). At the same time, the efficiency of canopy interception depends strongly on, but is not limited to, the effects of air temperature and wind speed (Cebulski and

Pomeroy, 2025; Katsushima et al., 2023; Lundquist et al., 2021). Next, melting processes are primarily driven by incoming shortwave radiation to the snow surface, which is higher in the open and within sparser canopies (Mazzotti et al., 2019b). The effects of shortwave radiation on canopy heating and emitted longwave radiation are also more pronounced in sparse canopies exposed to direct shortwave radiation (Haagmans et al., 2025; Webster et al., 2017). The strengths of these radiative differences are subsequently more pronounced in mid-latitude

compared to higher-latitude environments (Seyednasrollah and Kumar, 2019).

Apart from prevailing climatic conditions, variability in meteorological conditions from one winter to the next also affects the evolution of (forest) snow cover patterns. This interannual variability affects snow accumulation and ablation processes differently in forests compared to open areas, with the most substantial differences in snow storage between forested and open sites typically observed during warm and wet winters (López-Moreno and

Stähli, 2008). In contrast, these differences are usually smaller during cold and dry winters (Strasser et al., 2011). Meanwhile, the variability of forest snow cover increases during prolonged dry spells (López-Moreno and Latron, 2008) and cold, snow-scarce winters (López-Moreno and Stähli, 2008). Inter-annual variability of meteorological conditions has also been key in explaining snow accumulation and ablation dynamics across opposing forested slopes of a sub-alpine valley (Mazzotti et al., 2023).

Finally, topographic controls, particularly elevation and aspect, are additional key factors contributing to the complex differential patterns of snow storage between forests and open areas. At low and mid-elevations, snow can be deeper and last longer in the open compared to adjacent forest sites, whereas forests at higher elevations can maintain snow longer than the adjacent open site (Roth and Nolin, 2017). Topographic aspect has also been observed to have strong controls on differences in snow retention between open and forested sites. Forests on sun-

exposed slopes have been found to reduce incoming radiation to the snow surface and delay snowmelt timing compared to adjacent open sites, whereas forests on the opposing shaded slope increase radiation and advance the timing of snowmelt relative to the open site (Ellis et al., 2011). Moreover, consistently deeper snow depths were observed on a forested, shaded slope compared to the opposing sun-exposed forested slope (Koutantou et al., 2022). A detailed modelling study on these opposing sunlit and shaded slopes showed that forest structure, year-

to-year weather variability, and topography interact in intricate ways to form complex accumulation and ablation patterns of forest snow (Mazzotti et al., 2023). This is particularly relevant to consider for regions with transitional climates (Dickerson-Lange et al., 2023), such as the European Alps (Bozzoli et al., 2024).

Many forest snow studies have investigated only one or two of the key controls in isolation (i.e., climate, forest structure, year-to-year weather variability, and/or topography) as scale or data availability limited the ability to

disentangle how the four key controls interact across landscapes with complex topography (e.g., Ellis et al., 2011; Roth and Nolin, 2017; Seyednasrollah and Kumar, 2019). Especially in the European Alps, studies have been limited to small study sites and did not extend beyond the scale of small Alpine valleys (e.g., Geissler et al., 2023; Mazzotti et al., 2023). The spatiotemporal effects of montane forests on snow water resources, resulting from the combined effects of forest structure, topography, and year-to-year weather variability in different climate zones in

Europe, are thus still largely unknown. As snow storage in the European Alps declines (Marty et al., 2017; Matiu et al., 2021), it is becoming increasingly important to understand the different controls on snow water storage across the European Alpine range to predict, manage, and understand the consequences of such declines. Even more so, since past land-use-driven human interferences, as well as natural disturbances such as bark beetles, wildfires, and windthrow continue to strongly influence forest structure in the European Alps (Bebi et al., 2017).

Past studies of combined controls on forest snow storage have mainly been limited by the lack of spatiotemporally continuous forest snow observations in complex landscapes over the spatial extents required to thoroughly investigate interactions between climate, topography, interannual weather patterns, and forest structure. While airborne laser scanning (ALS) has been employed for repetitive mapping of montane forest snow cover in complex terrain (e.g., Broxton et al., 2015; Currier et al., 2019; Kostadinov et al., 2019; Mazzotti et al., 2019a; Safa et al.,

2021), continuous mapping across entire mountain ranges has yet to be achieved. Where comprehensive observational datasets are missing, advanced process-based models can complement our understanding of forest snow processes (Harpold et al., 2020; Mazzotti et al., 2023). These models have the advantage of providing consistent gapless multi-variable datasets that are impossible to observe over entire mountain ranges.

In this study, we use output from the FSM2oshd model, which is run by the national operational snow hydrological service (OSHD) across Switzerland and adjacent river basins, covering the central European Alps (Mott et al.,



2023). FSM2oshd is a mass and energy balance-based forest snow model that was specifically developed to run over large spatial scales while maintaining process representation and sub-grid parameterization for snow-canopy-atmosphere interactions. A point-based version of the model (FSM2) was used in Mazzotti et al. (2023) at 2 m spatial resolution across six winters to explore interactions between forest structure, topography, and weather in a small Swiss alpine catchment. Upscaling approaches for running FSM2 as the gridded FSM2oshd have been validated in Mazzotti et al. (2021), making it a suitable model choice for this study's purpose. Model output for eight consecutive hydrological years (2017–2024) is used to answer the following research questions (RQ):

*RQ1: What is the overall impact of forests on snow storage in the central European Alps?*

*RQ2: How do topography and climate across the study region affect forest impacts on seasonal snow dynamics?*

*RQ3: How do forest impacts on seasonal snow dynamics vary between years?*

First, the capability of the FSM2oshd model framework to accurately represent forest snow cover dynamics at its 250 m resolution is demonstrated in six focus regions. Subsequently, the FSM2oshd output is analyzed and discussed with respect to the above research questions RQ1–RQ3, yielding new insights on forest snow dynamics and its driving factors across large spatiotemporal scales. To the best of our knowledge, such an in-depth analysis of the current state of forest snow water resources over the central European Alps has not been attempted to date.

## 2 Methods

### 2.1 Study area

The study area is situated in central Europe, slightly extending across the Swiss national borders to include the tributaries to all lakes and rivers in Switzerland (Fig. 1). It spans nearly 58,000 km², with elevations ranging from 184 m to 4,806 m above sea level. 48.5 % of the area is above 1,000 m, and 19.4 % is above 2,000 m. The climate on the north side of the Alps is strongly influenced by its proximity to the Atlantic Ocean, whereas the Mediterranean Sea primarily influences the south side of the Alps (MeteoSwiss, 2025a).

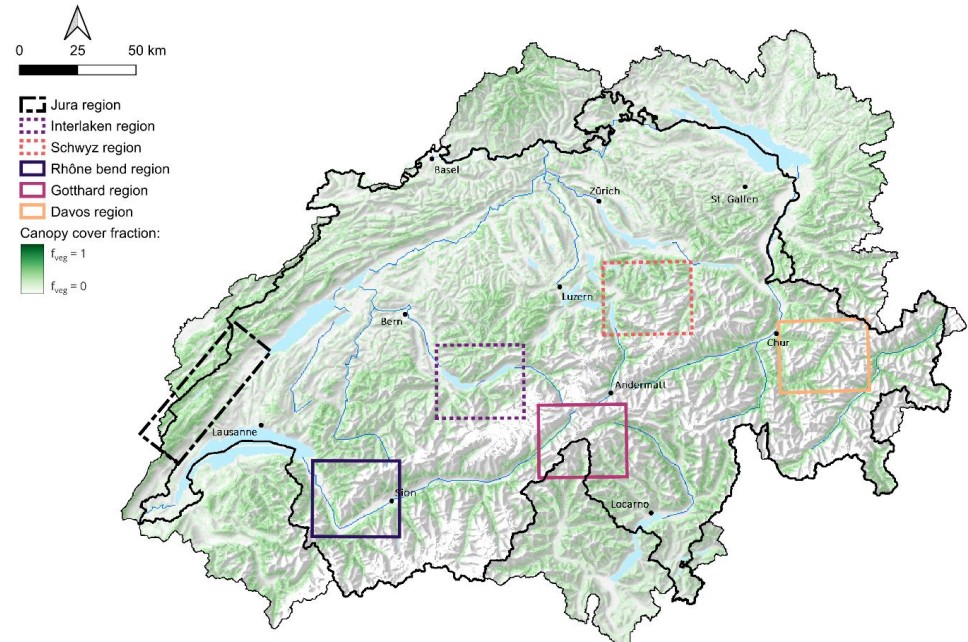

**Figure 1: Map of the study area with canopy cover fraction ($f_{veg}$) and six focus regions defined to analyze forest snow cover across different climatological conditions and forest types.**

Across the study area, 41 % is forested or directly affected by its nearby presence (e.g., shading). Therein, six focus regions, each approximately 1,200 km² in size, were defined to analyze forest snow cover dynamics across different climatological conditions and forest types (Fig. 1). These are grouped as follows: the non-Alpine Jura region and the northern Alpine flank (Interlaken, Schwyz regions), which are generally exposed to weather systems



originating from the Atlantic, and the more central Alpine regions (Rhone bend, Gotthard, Davos regions) that are exposed to weather systems originating in the Atlantic and the Mediterranean. Characteristics of each focus region are summarized in Table 1.

**Table 1: Topographical, land cover, and meteorological characteristics of the six focus regions outlined in Fig. 1. Meteorological characteristics are provided as an average over grid cells with forest land cover during the study period.**

|  | Jura (A) | Interlaken (B) | Schwyz (C) | Rhone bend (D) | Gotthard (E) | Davos (F) |
|---|---|---|---|---|---|---|
| Elevation range (m a.s.l.) | 430–1660 m | 556–4007 m | 418–3049 m | 347–3236 m | 408–3416 m | 430–3309 m |
| Area below 2000 m | 100 % | 84.1 % | 88.9 % | 71.8 % | 35 % | 43.4 % |
| Forest cover fraction | 21.8 % | 45.2 % | 20.7 % | 26.2 % | 29 % | 34.4 % |
| Mean leaf area index (LAI) | 2.46 | 1.75 | 2.28 | 2.05 | 1.71 | 2.65 |
| Mean canopy cover fraction of forested terrain ($f_{veg}$) | 0.53 | 0.41 | 0.47 | 0.46 | 0.42 | 0.53 |
| Mean $T_{annual}$ in forested terrain | 8.33 °C | 8.03 °C | 7.99 °C | 8.16 °C | 5.9 °C | 5.63 °C |
| Mean $T_{DJF}$ in forested terrain | 0.69 °C | 0.83 °C | 0.70 °C | 0.79 °C | -1.22 °C | -1.53 °C |
| Mean annual precipitation in forested terrain | 1263 mm | 1293 mm | 1666 mm | 1042 mm | 1427 mm | 1115 mm |

### 2.2 The Swiss operational snow-hydrological model framework

Based on Mott et al. (2023), we provide a brief overview of the OSHD model framework, focusing on the FSM2oshd snowpack model and its forest snow routines. FSM2oshd provides physics-based snow and melt distribution simulations at 250 m resolution over the entire study area. It is forced by output from the 1 km numerical weather prediction system COSMO, run by MeteoSwiss (2025b), which is downscaled to 250 m. Moreover, assimilation of data from 444 snow monitoring stations in the study area ensures consistency with available observations of snow height and SWE using methods detailed in Magnusson et al. (2017, 2014) and Cluzet et al. (2024).

The snow cover's mass and energy balances are solved using separate model instances for open, forested, and glacierized areas, while snow is not accumulated on large water bodies that never freeze. Accordingly, grid cells are divided into four corresponding land cover fractions (cf. Fig. 2, Mott et al., 2023), and grid cell-level snow cover properties are computed as weighted averages of the different instances. A forest mask is used to partition grid cells into open areas and forest-covered fractions. Note that open areas influenced by the presence of adjacent forest, i.e., within 20 m distance of forest edges, are included in the forest fraction (see Fig. 2.2–2.3). All key processes through which the forest canopy affects mass and energy exchange between the atmosphere and the sub-canopy snowpack are incorporated in FSM2oshd simulations: canopy interception of snowfall is followed by either sublimation into the atmosphere or unloading onto the ground. The canopy also regulates the transmission of shortwave radiation while enhancing longwave radiation and attenuating wind. Accurate representation of forest snow processes at the 250 m model resolution is achieved by using canopy structure descriptors (e.g., canopy cover fraction $f_{veg}$) and time-varying transmissivities computed at high spatial resolution, which are subsequently averaged over the forest fraction of a grid cell (Mazzotti et al., 2021). For example, shortwave transmission is calculated at a 10 m spacing to account for the effect of heterogeneous canopy structure; the results are then upscaled to the model grid resolution, as described in Mazzotti et al. (2021) and Webster et al. (2023). Note that the model instance representing forest snow only runs for grid cells that have forest cover, while simulations for open (i.e., non-forested) conditions are calculated for every grid cell. This approach enables a direct assessment of the effect of forest cover relative to open conditions, even for grid cells that are fully covered by forest.

### 2.3 Evaluating modeled forest snow cover dynamics using PlanetScope RGB imagery

The FSM2 model has been extensively validated, particularly with respect to individual forest snow processes (Mazzotti et al., 2020a), hyper-resolution simulations (Mazzotti et al., 2020b, 2023), and its performance in upscaled simulations from 1 m to 50 m (Mazzotti et al., 2021). In this study, we also demonstrate the capabilities of FSM2 at the 250 m resolution (i.e., FSM2oshd) to accurately model forest snow cover across the range of climates and forests in the study area.





Over the European Alps, snow cover retrievals derived from high-resolution (1–5 m) optical satellite imagery are the only distributed and regularly available information for evaluating modeled forest snow over large areas and at scales where forest snow processes vary. The PlanetScope constellation acquires such satellite imagery, providing near-daily coverage of the six focus regions at approximately 3 m resolution (Frazier and Hemingway, 2021). Because PlanetScope acquisitions are collected near the edge of very low Earth orbit at small off-nadir view angles, the resulting images enable direct peering into complex forested terrain, including small forest gaps (Pflug et al., 2024). This study utilized PlanetScope acquisitions, available as image composites, to assess modeled forest snow cover in the focus regions. The composites were selected based on the following criteria: i) no or minimal cloud cover existed over the evaluated region, ii) no intercepted snow was stored in forest canopies, and iii) alignment with three specific evaluation goals related to snow cover dynamics:

- Inter-annual evaluation: one region, one PlanetScope composite per year acquired around April 1st, over six years
- Seasonal evaluation: one region, six PlanetScope composites within one year
- Inter-regional evaluation: six regions, one PlanetScope composite for approximately the same date (around April 1st).

The data product used in this study is 'ortho_visual', which is post-processed explicitly for visual analyses (Planet Labs Inc., 2023). Further details are provided in Table 2.

**Table 2: An overview of the analyzed PlanetScope composites for the focus regions and respective evaluation goals. The number of evaluated grid cells per land cover type is provided for each PlanetScope composite of a focus region. The acquisition time of the images in the composites is between 9 and 11 UTC.**

| Focus region | Date | Evaluation goal | No. evaluated grid cells with open area land cover | No. evaluated grid cells with sparse forest land cover | No. evaluated grid cells with dense forest land cover |
|---|---|---|---|---|---|
| Rhone bend | 2020-04-01 | Inter-regional | 773 | 748 | 625 |
| Interlaken | 2020-04-01 | Inter-regional | 748 | 1030 | 613 |
| Gotthard | 2020-04-03 | Inter-regional | 564 | 595 | 373 |
| Schwyz | 2020-04-01 | Interregional | 743 | 616 | 505 |
| Jura | 2022-02-10 | Inter-regional | 636 | 819 | 670 |
| Davos | 2019-03-29 | Inter-annual | 529 | 648 | 576 |
| Davos | 2020-02-07 | Seasonal | 270 | 268 | 272 |
| Davos | 2020-03-08 | Seasonal | 454 | 536 | 503 |
| Davos | 2020-03-19 | Seasonal | 489 | 525 | 515 |
| Davos | 2020-04-01 | Seasonal, inter-regional, inter-annual | 652 | 510 | 503 |
| Davos | 2020-04-13 | Seasonal | 652 | 490 | 578 |
| Davos | 2020-04-23 | Seasonal | 789 | 503 | 511 |
| Davos | 2021-04-01 | Inter-annual | 1008 | 887 | 897 |
| Davos | 2022-03-28 | Inter-annual | 693 | 678 | 708 |
| Davos | 2023-04-06 | Inter-annual | 854 | 826 | 789 |
| Davos | 2024-04-06 | Inter-annual | 772 | 734 | 596 |

The procedure for evaluating modeled snow cover in FSM2oshd consisted of six steps, the initial four of which are visualized in Fig. 2.

**Step 1: Select a PlanetScope composite for evaluation**
For a specific evaluation goal, we selected a suitable PlanetScope composite, ensuring satisfactory quality through visual inspection. The corresponding composite was downloaded, and the 250 m FSM2oshd model grid overlaid on it (Fig. 2.1).

**Step 2: Assign land cover types**
Based on land cover information available within the FSM2oshd framework, for each model grid cell in a focus region, we determined whether open areas and/or forests are present. When open areas are present (Fig. 2.2, gray area, inverse of forest mask), the grid cell was assigned to the open land cover mask. If forest cover is present (Fig. 2.2, green area, forest mask), the grid cell was assigned either to the sparse or the dense forest land cover mask based on canopy cover fraction as a proxy of forest cover density (sparse: $f_{veg} < 0.5$, dense: $f_{veg} > 0.5$, see Fig. 1). Note that this procedure led to double assignments wherever open land cover and either sparse forest or dense forest co-occur in a grid cell (cf. Fig. 2, Mott et al., 2023).



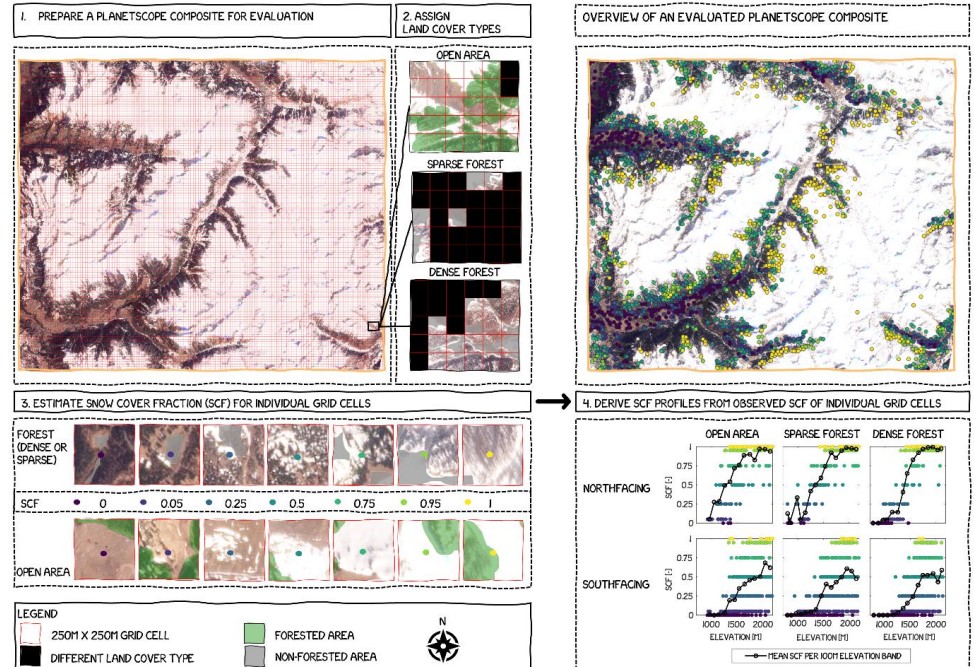

**Figure 2: The initial four (of six) steps in the workflow for validating modeled (forest) snow cover fraction (SCF) using PlanetScope RGB imagery. The PlanetScope composite captures the Davos region.**

**Step 3: Estimate snow cover fraction (SCF) for individual grid cells**

To the best of our knowledge, no algorithm is currently available to reliably estimate within-forest SCF in PlanetScope retrievals (e.g., in small forest discontinuities). We therefore conducted a manual assessment of SCF. Based on the observable snow cover fraction in a grid cell, a SCF estimate was made by assigning a corresponding SCF class*:* bare (0), very little (0.05), patchy (0.25), about half (0.5), mostly continuous (0.75), almost continuous (0.95), and continuous (1); see Fig. 2.3. Note that the SCF of grid cells was assessed separately for the different land cover types, and only for that part of a grid cell assigned to the respective land cover type.

Using seven distinct SCF classes was the best trade-off between covering a range (0-1) of SCFs (i.e., information content) and reliable manual assignment. On average, per PlanetScope composite, SCF was assessed for 630 grid cells for each of the three land cover types, ensuring representation of the entire SCF range (0–1), and varying elevations and aspects; see Fig. 2.4 (upper panel). In total, 16 composites and ~30.000 individual grid cells were evaluated (Table 2).

**Step 4: Derive SCF profiles from the observed SCF of individual grid cells**

Estimated SCFs of individual grid cells were binned into 100 m elevation bands for each land cover type to provide a mean SCF value per elevation band while also accounting for aspect (i.e., north-facing ($> 270° – \leq 90°$) or south-facing ($> 90° – \leq 270°$). A PlanetScope composite hence yielded six SCF profiles (see Fig. 2.4, lower panel).

**Step 5: Derive SCF profiles from the modeled SCF of individual grid cells**

Here, step 4 was repeated with model data to yield equivalent profiles of modeled SCF for those grid cells assessed in step 3.

**Step 6: Compare observed SCF to modeled SCF**

Finally, the SCF profiles enabled the evaluation of differences between modeled and observed SCF dynamics, separately for each land cover type and according to aspect.

**2.4    Metrics for analyzing modeled forest impact on snow cover dynamics**

To isolate the impact of forest on snow cover dynamics in each grid cell and compare it with non-forested conditions, our approach leveraged the different model instances used in the FSM2oshd framework to represent different land cover types (cf. Sect. 2.2). Several metrics were computed to characterize the snow season for each



of these two model outputs, as well as for the grid-cell-averaged product accounting for all land cover types. Metrics were calculated over all grid cells and eight hydrological years (HY) from 2017 to 2024, each starting on September 1st.

First, a qualitative, categorical characterization of the snow season across a region is offered by snow cover types. Here, we assessed these for all three model outputs and across the whole study area using a simplified classification solely relying on seasonal snow cover duration, adapted from Sturm et al. (1995), Sturm and Liston (2021), and López-Moreno et al. (2024):

- *Ephemeral snow*: a snow cover that persists for more than one but less than 60 consecutive days.
- *Marginal snow:* a snow cover that persists during 60-120 consecutive days
- *Seasonal snow*: a snow cover that persists for at least 120 consecutive days but has disappeared by August 31st (end of HY).
- *Perennial snow*: snow cover accumulated over the season persists until August 31st, i.e., through the hydrological year and beyond.

Second, we relied on quantitative snow accumulation and melt descriptors for further in-depth analysis. These follow Mazzotti et al. (2023) and include:
- *Peak SWE*: the maximum value of snow water equivalent on the ground.
- *Snow disappearance date (SDD)*: the date with the first occurrence of SWE < 10 mm after peak SWE, also referred to as melt-out.
- *Center time of snowmelt runoff (CT):* the date at which 50 % of the total annual snowmelt runoff (at the grid-cell level) has been released (Stewart et al., 2004).

Unless specifically stated, the analyses in the results section assessed the impact of forest cover in terms of the difference between the model instances representing forested and open land cover types.

## 3  Results

### 3.1  Evaluation of modeled forest snow cover based on PlanetScope imagery

Figure 3 shows an evaluation of modeled snow cover fraction during spring for six consecutive years in the Davos region. Aspect-dependent SCF differences are apparent. Approximating the snowline by SCF = 0.5, we found an elevation difference between the snowlines on north- and south-facing aspects of approximately 400 m during this time of the snow season. This elevation difference was similar between PlanetScope acquisitions from different years. However, the absolute snowline elevation varied from year to year, despite observations made at the same time of year. For example, in 2019 and 2021, the snowline was 400 m lower than in 2023 and 2024. Remarkably, differences in SCF profiles between the three land cover types (open area, sparse forest, and dense forest) remained typically within ~50 m, though slightly greater in some years (2023, 2024) than in others (2019, 2021). All the above findings were generally well represented by the FSM2oshd model, even though in some cases the model failed to replicate the slight differences between SCF profiles of the three land cover classes.

Figure A1 illustrates the progression of the snowline with elevation throughout winter and spring of 2020 in the Davos region, representing the inter-seasonal evaluation. Both modeled and observed SCF profiles showed a progressive increase of ~500 m in snow line elevation between March 8th and April 23rd, while the spread between aspects gradually increased. FSM2oshd could even represent specific features, such as the observed substantial SCF difference between sparse and dense forests in north-facing slopes on March 19th, or the drop in SCF at high elevation in south-facing slopes on April 14th and April 23rd. Considerable discrepancies were only present in the comparison for February 7th, when the modeled forest SCF was underestimated shortly after a minor snowfall two days before the PlanetScope scene acquisition, due to melting starting too early in the model.





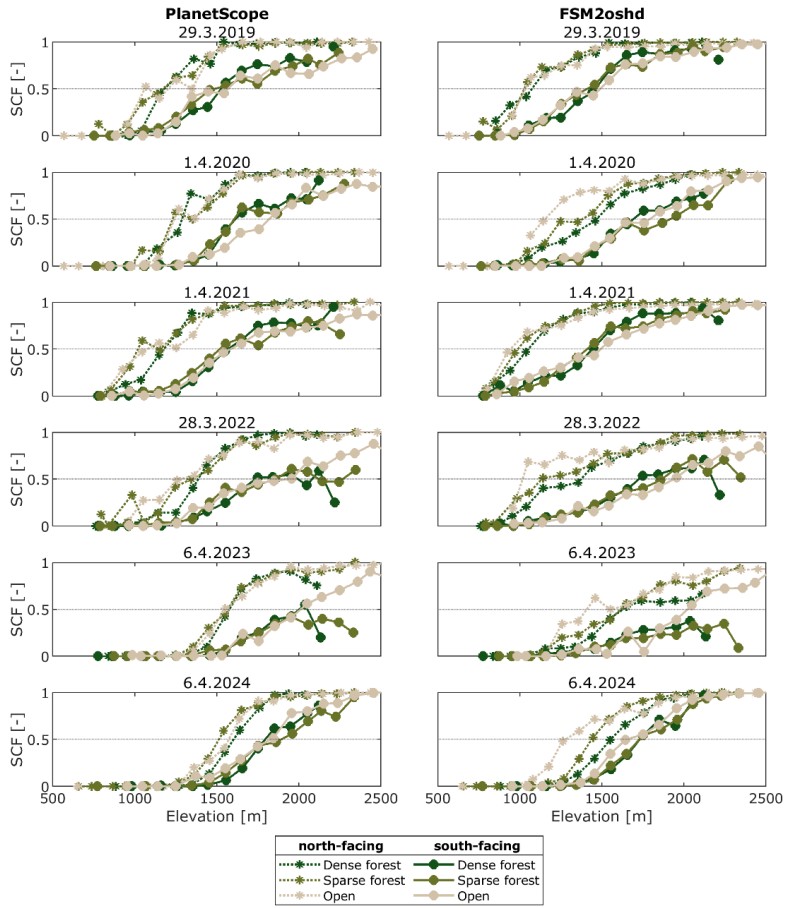

**Figure 3: Comparison of observed (PlanetScope) and modeled (FSM2oshd) snow cover fraction (SCF) for the Davos focus region and the 2019–2024 period. SCF profiles are shown separately for three land cover types (i.e., dense forest, sparse forest, and open area) on north- or south-facing slopes.**

The interregional evaluation of SCF (see Fig. A2) showed similar patterns across the six focus regions to those observed in the Davos region. The observed interregional differences were minor compared to the variability between aspects and years (cf. Fig. 3). Yet, differences of SCF profiles between aspects were notably smaller in the Rhone bend region compared to the Gotthard region. FSM2oshd provided an excellent match to SCF observations for the Rhone bend, Gotthard, and Davos regions, but less so for the Interlaken and Schwyz regions. Lower accuracy in these two cases could also be attributed to a minor snowfall that occurred two days before the PlanetScope image was acquired. This snowfall of a few centimeters was sufficient to make the snow cover appear near-uniform in the observations, but FSM2oshd did not accurately represent this. The Jura region stands out in that the profiles of observed SCF in south-facing aspects were captured very well by FSM2oshd, whereas modeled SCF on north-facing aspects at elevations of 700 m - 1100 m was significantly lower than the observed values. This again resulted from a prior snowfall event, where the resulting snow cover persisted longer on the steep, shaded north-facing slopes of the Jura than what was modeled.

Mean absolute error (MAE) statistics further confirmed the excellent match between observations and model (see Table A1). The MAEs, averaging at 0.16 SCF, were low and remarkably consistent across PlanetScope composites and land cover types. Slightly larger MAEs (~0.25) were only determined for the evaluations of the Interlaken and Schwyz regions on April 1, 2020. Note that this assessment was based on the evaluation of many grid cells for each land cover class within a focus region. On average across all PlanetScope composites, 4 % of all grid cells classified as 'open area,' 13 % of all grid cells with a 'sparse forest,' and 15 % of all grid cells with a 'dense forest'




land cover type were evaluated. We thus concluded that FSM2ohsd is well capable of representing observed SCF variability between years, aspects, and throughout a winter season, even if in 3 of the 16 composites, some shortcomings of the model's response to recent snowfall were notable. Therefore, we consider the model results to provide a highly accurate representation of snow cover evolution in both open and forested environments, which 285 constitutes the basis for the following analysis.

### 3.2 Overall impact of forests on seasonal snow cover

Figure 4 summarizes the impact of forests on snow water resources across the study area and for the 2017–2024 hydrological years. The maximum volume of water stored as snow during a season averaged at 7.0 km³ and ranged from 4.8 km³ to 10.6 km³ (Fig. 4A), corresponding to a mean peak SWE between 87 mm and 192 mm. In the 290 forested part of the study area alone, peak SWE storage reached a maximum of 2.6 km³, with an average of 1.3 km³, and a minimum of 0.7 km³ (Fig. 4B). Forest peak SWE typically occurred 1-2 months earlier than total peak SWE.

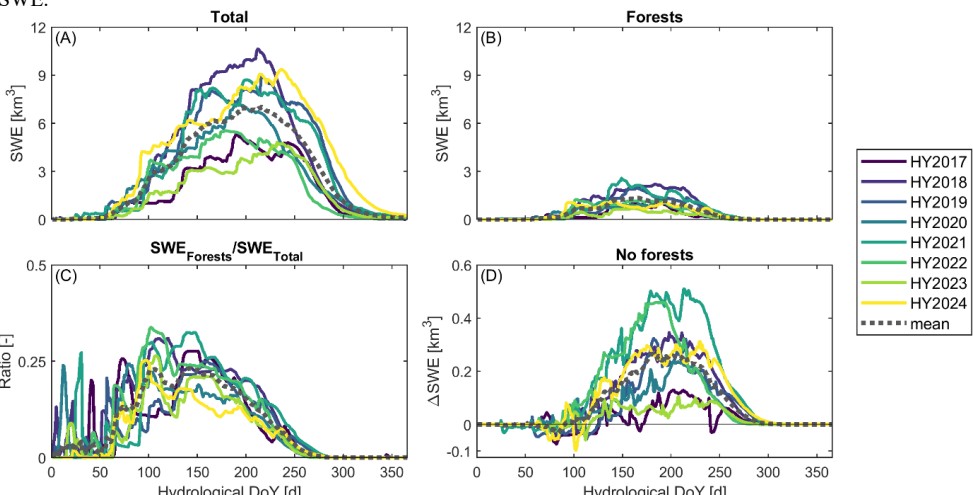

**Figure 4: Snow storage evolution during eight hydrological years (HY) for each day of the year (DoY) in terms of snow water equivalent (SWE), summed across the whole study area (A) and for the forested part of the study domain only (B). Ratio of SWE stored in forests to total SWE storage (C), and storage difference if there were no forests (D).**

To quantify the relevance of forest snow water storage, we computed the fraction of SWE stored in forests relative 295 to total SWE (Fig. 4C). This fraction reached an absolute maximum of 0.34 in early winter of December 2021. On average, it peaked at 0.23 in early winter. It then remained relatively stable for about two months before decreasing as total SWE storage continued to increase, whilst forest SWE storage remained approximately constant. The period during which forests stored snow was significantly shorter than the overall SWE storage period in the study area. This difference resulted from the abundance of snow at elevations well above the tree line.

Alternatively, comparing actual snow water storage to a model scenario without forests provided an estimate of snow water storage sensitivity to forest cover in absolute terms (Fig. 4D). If forest cover was unaccounted for over the entire model domain, total SWE storage would, at its peak, increase by only 0.25 km³ on average, which is 3.7% of the current mean total peak SWE (cf. Fig. 4A and Fig. 4D). The maximum SWE increase in this scenario reached 0.5 km³ during HY2021 and occurred 11 days later than the total peak SWE during that year (Fig. 4A). 305 This indicates the potential for prolonged overall storage without trees, as can also be observed by comparing the tails of the SWE curves for forests (Fig. 4B) and no forests (Fig. 4D), respectively.

The impacts of forests on snow cover dynamics outlined above mean that the snow cover type attributed to a specific location can also differ, depending on whether forest cover is present or not. Using the snow cover type classification detailed in Sect. 2.4, we present a qualitative assessment of forest impacts on seasonal snow in Fig. 310 5. Over 95 % of the land area in the study domain was snow-covered during the study period, for which the dominant snow cover types were ephemeral (50.5 %) and seasonal (34.5 %), as illustrated in Fig. 5A. Ephemeral snow cover was primarily found in the lowlands below 600 m, while seasonal snow cover was typically present in most of the Alps above 1000 m. Marginal snow cover (14 %) mainly occurred above 750 m in the pre-Alps and high alpine valleys, acting as a transition zone between ephemeral and seasonal snow cover. Perennial snow cover





(1.5 %) closely aligned with glacier cover and was found only above 2500 m. While elevation was the primary
factor controlling snow cover type, it also depended on aspect (Fig. 5C); for example, seasonal snow tended to
occur approximately 200 m lower on north-facing aspects compared to south-facing aspects.

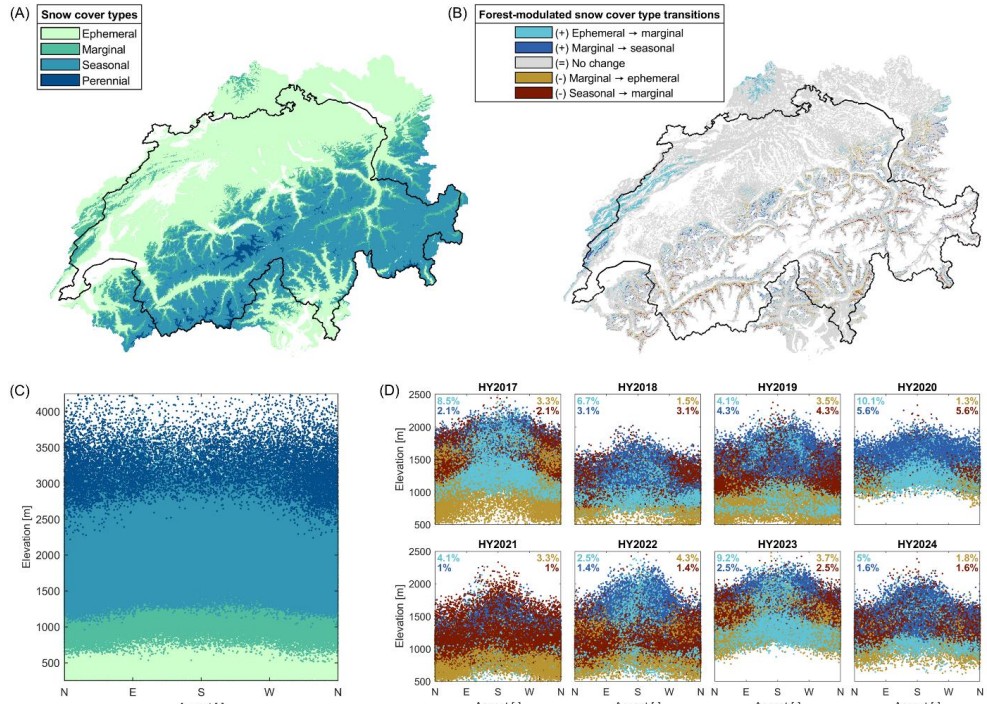

**Figure 5: Dominant snow cover types across the study area during the 2017–2024 hydrological years (A), for which (C) captures the distribution across elevation and aspect. Transitions between snow cover types due to the presence of forest are shown in (B), while (D) captures their variability across aspect and elevation for each year. The four percentage rates shown for each hydrological year indicate the ratio of grid cells that transitioned between snow cover types due to the presence of forest.**

Comparing the snow cover types determined for the model outputs of the forested and open land cover types revealed where and how forests, in addition to topographic factors, affected this classification. Transitions between

snow cover types due to the presence of forest are mapped out in Fig. 5B. Of all grid cells with forest cover and seasonal snow, 8.5 % had a snow cover duration that was extended by forests: 6 % were classified as marginal in the presence of forest and as ephemeral in its absence, while 2.5 % were seasonal with forest and marginal without. The opposite, i.e., a shortening of the snow cover season, was found on 4.3 % of these grid cells, with 2.5 % classified as ephemeral with forest and marginal without, and 1.8 % as marginal with and seasonal without forest,

respectively.

These transitions between snow cover types occurred across large areas in the Jura and the pre-Alps. Here, the snow cover type was identified as marginal in the presence of forest but as ephemeral in its absence. The same type of transition was also found across the Alps on some south-facing slopes above 1,000 m. Alpine south-facing aspects above 1500 m more often featured seasonal snow cover if forest was present, but marginal snow cover in

the open. On north-facing aspects above 1500 m, snow cover type was commonly marginal in the presence of forest but seasonal in the open. Below this elevation on north-facing aspects, snow cover types were ephemeral and marginal in the presence and absence of forest, respectively. However, considerable variability in snow cover type divergencies existed between hydrological years, as visualized in Fig. 5D.





### 3.3 Spatial differences in forest impacts on snow accumulation and ablation dynamics across the study area


To explore how the impacts of forests on snow cover vary in space across complex topography and climatic gradients, we analyzed how snow cover dynamics descriptors introduced in Sect. 2.4 varied in terms of elevation and aspect across the six focus regions introduced in Sect. 2.1.

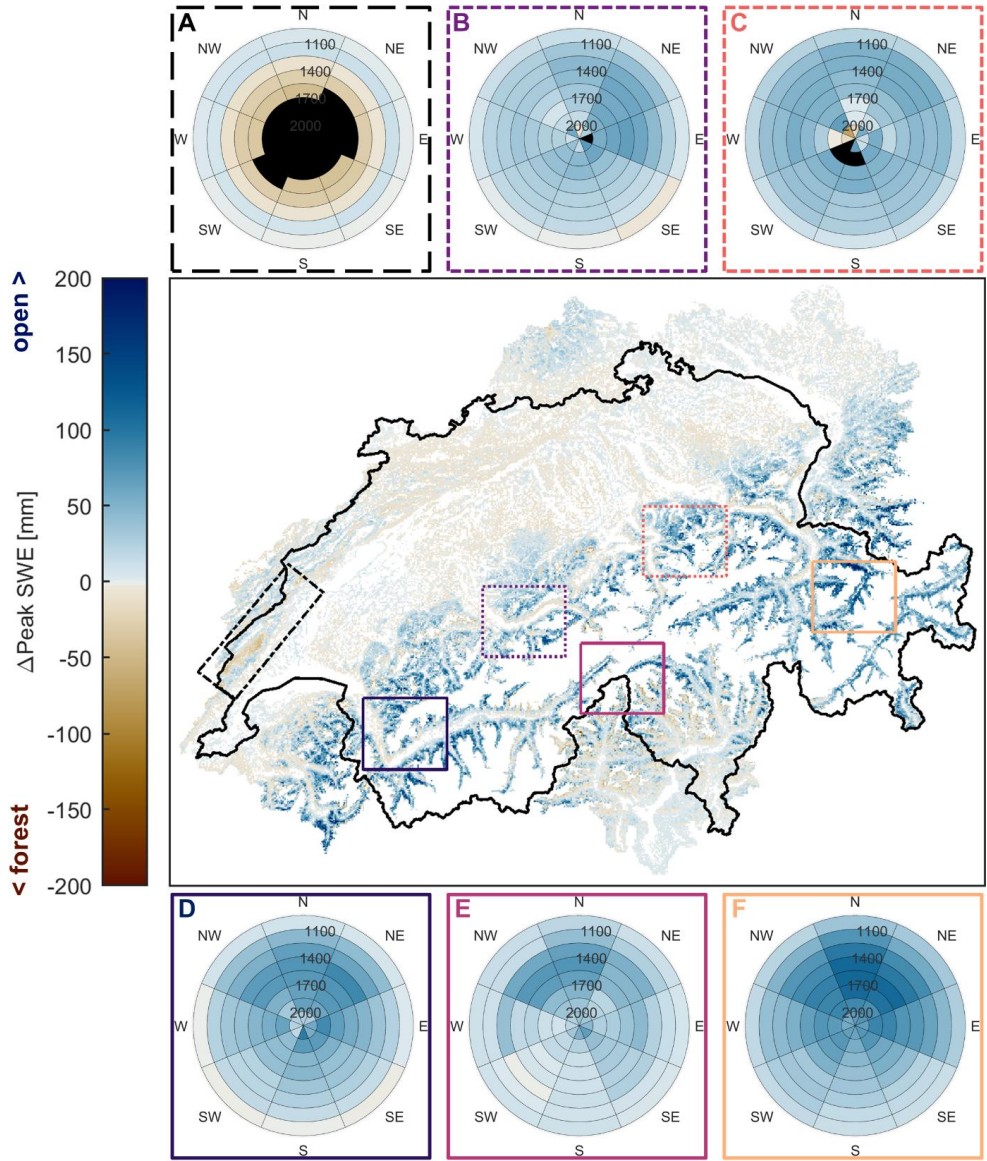

**Figure 6: Mean difference in peak SWE between open and forested areas (ΔPeak SWE) during the 2017–2024 hydrological years. For each of the six focus regions, a polar plot is provided (A–F). These show the average ΔPeak SWE for eight 150 m elevation bands (950–2150 m) across eight aspect classes, where the labels refer to the directions in which slopes face.**


Figure 6 presents the difference in peak SWE, i.e., ΔPeak SWE, computed as the peak SWE in the open land cover type minus the peak SWE in the forested land cover types. Note that the same sign convention was applied to all subsequent analyses. In the Alps, peak SWE in open areas was almost everywhere higher than in forested terrain.



Notably, a tendency for greater deficits in forests above ~1300 m existed. In contrast, at lower elevations and away from the Alps, peak SWE was slightly higher in forests. Moreover, the three more central Alpine focus regions exhibited aspect dependency, where peak SWE showed a remarkably higher deficit in forests on north-facing slopes (D-F, Fig. 6). Such aspect-dependent patterns were not observed in the two focus regions on the northern flanks of the Alps (B-C, Fig. 6). In the Jura Mountains (A), the highest elevations exhibited a distinct pattern. Here, forested areas stored slightly higher peak SWE compared to nearby open areas.

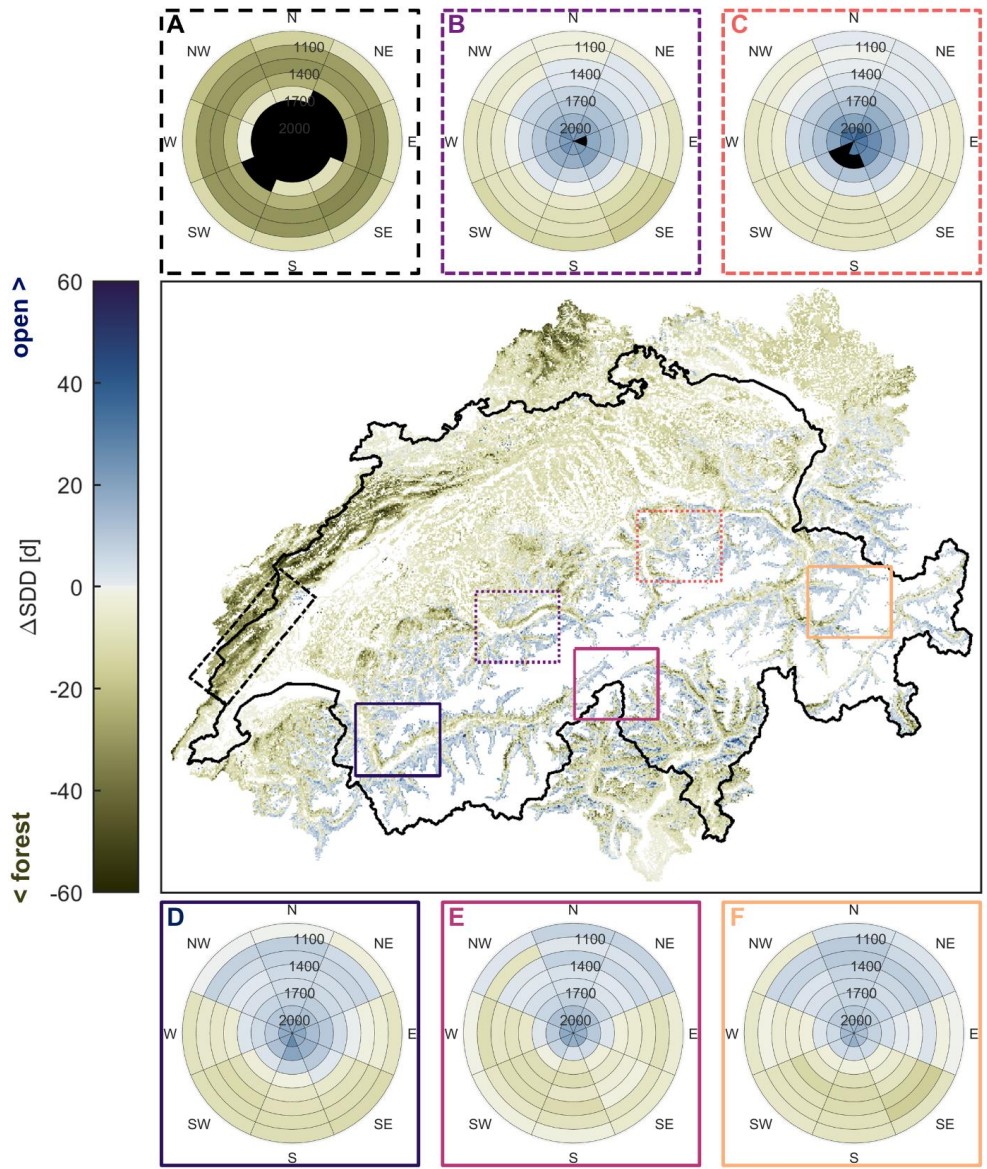

**Figure 7: Mean difference in snow disappearance day between open and forested areas (ΔSDD) during the 2017–2024 hydrological years. For each of the six focus regions, a polar plot is provided (A–F). These show the average ΔSDD for eight 150 m elevation bands (950–2150 m) across eight aspect classes, where the labels refer to the directions in which slopes face.**

Figure 7 presents the average difference in snow disappearance day between forested and open areas (ΔSDD). Across the Alps, whether snow persisted longer in forests or open areas was generally dictated by elevation and



aspect. Within the five Alpine focus regions (B-F, Fig. 7), snow persisted longer in open areas above ~1700–1850 m for 1 to 5 weeks, regardless of aspect. Below this elevation, snow remained longer in forests on south-facing slopes for a few days to two weeks. Oppositely, on north-facing slopes, snow persisted longer in open areas for a few days to two weeks. Within the (non-Alpine) Jura region, snow persisted longer in the forests for 1 to 4 weeks, regardless of aspect (A, Fig. 7). These differences in SDD were most pronounced in elevations between 1100 m and 1550 m.


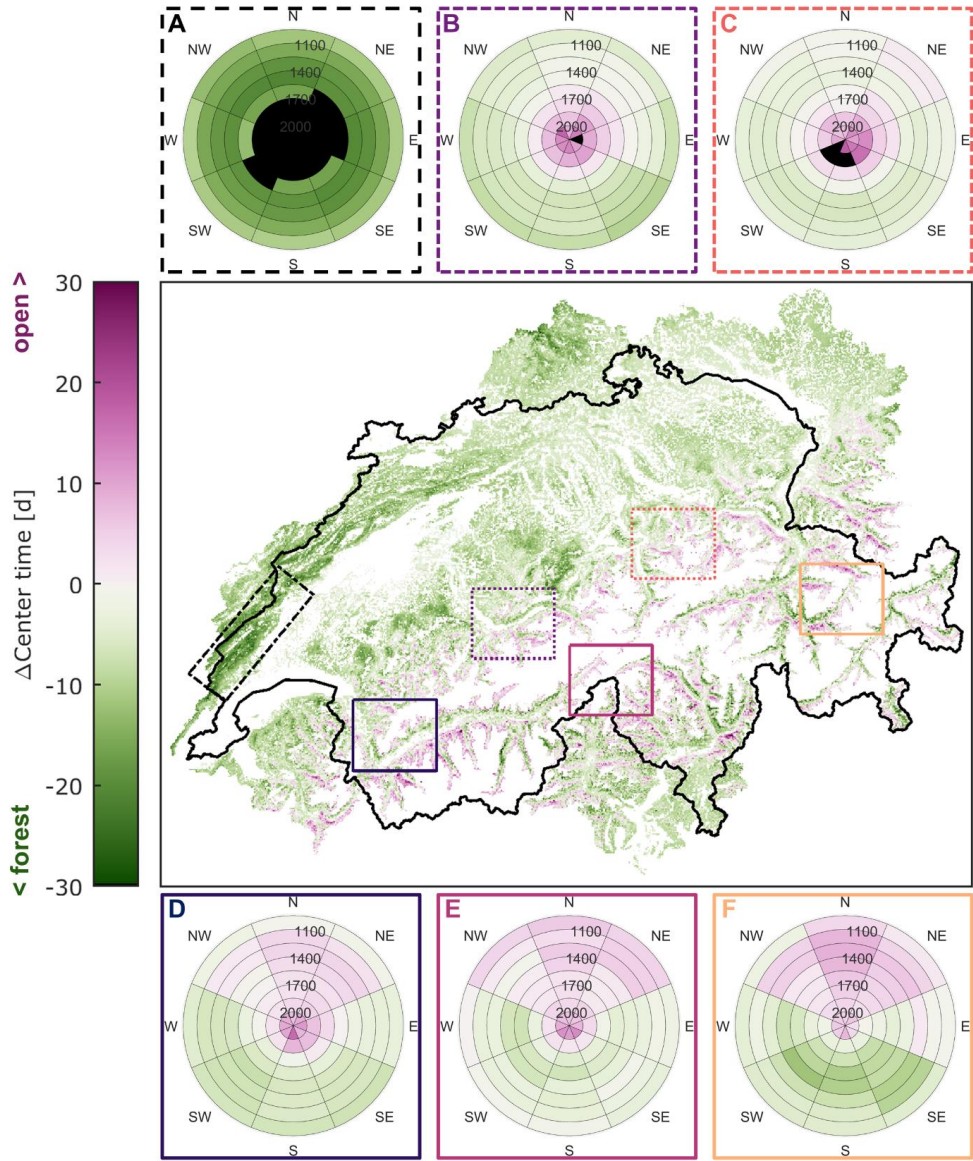

**Figure 8: Mean difference in the date on which 50 % of the snowmelt volume is released between open and forested areas (ΔCT) during the 2017–2024 hydrological years. For each of the six focus regions, a polar plot is provided (A–F). Polar plots show the ΔCT for eight 150 m elevation bands (950–2150 m) across eight aspect classes, where the labels refer to the directions in which slopes face.**

Much before the snow cover disappears, the snowpack already generates runoff from melting. The timing of snow accumulation relative to melting determines when precipitation, which is temporally stored as snow, becomes available for runoff generation. To contrast this timing between forested and nearby open areas, we analyze the





date on which 50 % of the snowmelt volume is released (CT) in Fig. 8. We found that, on average, whether CT
        occurred first in forested or nearby open areas strongly depended on the region, elevation, and aspect. Away from
        the tree line (<1850 m), the focus regions centered in the Alps (D-F, Fig. 8) exhibited a clear aspect dependence
        on whether CT occurred earlier in forests or in nearby open areas. Here, compared to open areas, forests on south-
        facing slopes delayed CT by at least a few days and up to two weeks. In contrast, forests on north-facing slopes
advanced CT by up to two weeks, which also applied more generally above 1850 m in all regions, regardless of
        aspect. Within the two focus regions on the northern Alpine flank (B-C, Fig. 8), forests (<1850 m) in all aspects
        delayed CT by at least a few days and up to two weeks, while these delays were more pronounced on south-facing
        slopes. Significantly longer delays of CT by forests of 2-3 weeks were observed in the (non-Alpine) Jura region
        (A, Fig. 8).

**3.4     Temporal variations in forest impacts on snow accumulation and melt dynamics**

        In this Section, we examined variations in forest snow cover dynamics over the eight hydrological years. The
        differences in peak SWE, SDD, and CT between forested and nearby open areas in the Davos region were analyzed.
        The winter conditions for each hydrological year were categorized as either snow-scarce (≤ -20 % mean peak
        SWE), average, or snow-rich (≥ +20 % mean peak SWE).

The difference in peak SWE storage between open areas and forests varied considerably between years and with
        elevation and aspect (Fig. 9A). Generally, peak SWE storage was higher in open areas, with this difference being
        more pronounced on north-facing slopes. Snow storage differences consistently increased with elevation up to the
        tree line. In snow-rich years, differences in peak SWE storage were amplified, while they were minimized during
        snow-scarce years. Notably, during the 2020 hydrological year, forests on south-facing slopes at lower elevations
had higher peak SWE storage.

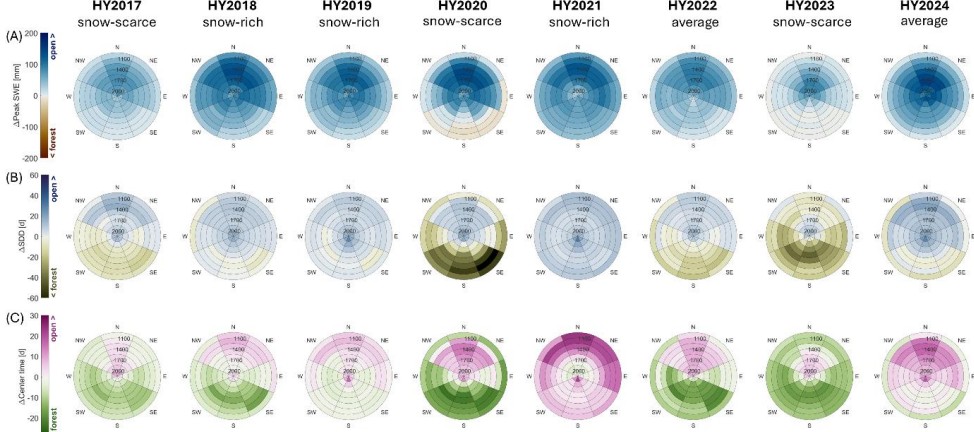

**Figure 9: Inter-annual variability of forest snow cover dynamics in the Davos region for eight hydrological years (HY),
characterized by differential peak SWE (A), differential snow disappearance day (B), and differential center time of
snow melt runoff (C) between forested and open areas. The winter conditions are categorized as either snow-scarce (≤ -
20 % mean peak SWE), average, or snow-rich (≥ +20 % mean peak SWE). Polar plots show the respective variable for
eight elevation bands, each spanning 150m (950–2150 m) across eight aspect classes, representing the directions in which
slopes face.**

The day of snow disappearance also exhibited considerable variability across elevations and aspects between years
(Fig. 9B). In many locations, snow persisted longer in forests in one year but disappeared earlier in another. This
variability in ΔSDD could be attributed to winter conditions: during snow-rich winters, snow generally persisted
longer in the open, while during average and snow-scarce winters, aspect-dependent patterns emerged as snow
persisted longer in south-facing forests. Near the tree line (>1850 m), ΔSDD was less pronounced, and during
        most winters, snow persisted longer in the open.

Differential patterns of the center time of snow melt runoff (ΔCT) showed additional variability that did not
necessarily follow patterns of ΔPeak SWE and ΔSDD linked to winter conditions (Fig. 9C). During snow-rich
winters, CT occurred later in forests on south-facing slopes for one year (HY2018), while occurring later in the
open during another year (HY2021). Conversely, during snow-scarce and average winters, patterns of ΔSDD and



ΔCT showed greater conformity; for example, snow persisting longer in forests generally co-occurred with CT occurring later in forests.

## 4    Discussion

### 4.1    The role of accumulation versus ablation for differential snow cover dynamics

The analysis presented in Sect. 3.3 reveals considerable differences between open and forested terrain in both peak SWE (Fig. 6) and SDD (Fig. 7). The question arises as to whether these differences occur primarily due to accumulation or ablation processes. On south-facing slopes in the Alps, snow typically persists longer in forests than in the open (B-F, Fig. 7), even though peak SWE is on average lower in these forests (B–F, Fig. 6). Hence, snow persistence in south-facing slopes appears to be mainly a result of slower ablation rates in forests in
comparison to the open. However, on north-facing slopes, snow persists longer in the open where peak SWE storage is also higher. Here, it is likely the difference in accumulation that drives the overall effect of forests on snow persistence.

However, spatiotemporal differences in accumulation and ablation, as well as their variations between forested and open terrain, are more intricate than the above description of the overall effect suggests. Figure 10 presents an
evaluation of the contrasting conditions between south and north-facing grid cells. While these plots only show grid cells from one region (Davos) and one elevation band (1400–1550 m), there is considerable variability in the amount of snow that builds up during the accumulation period in the open (peak $SWE_{open}$). This is primarily due to differences in concurrent melt, modulated by terrain shading and variable weather conditions. Spatial variability in snowfall is less influential in this regional evaluation (data not shown).

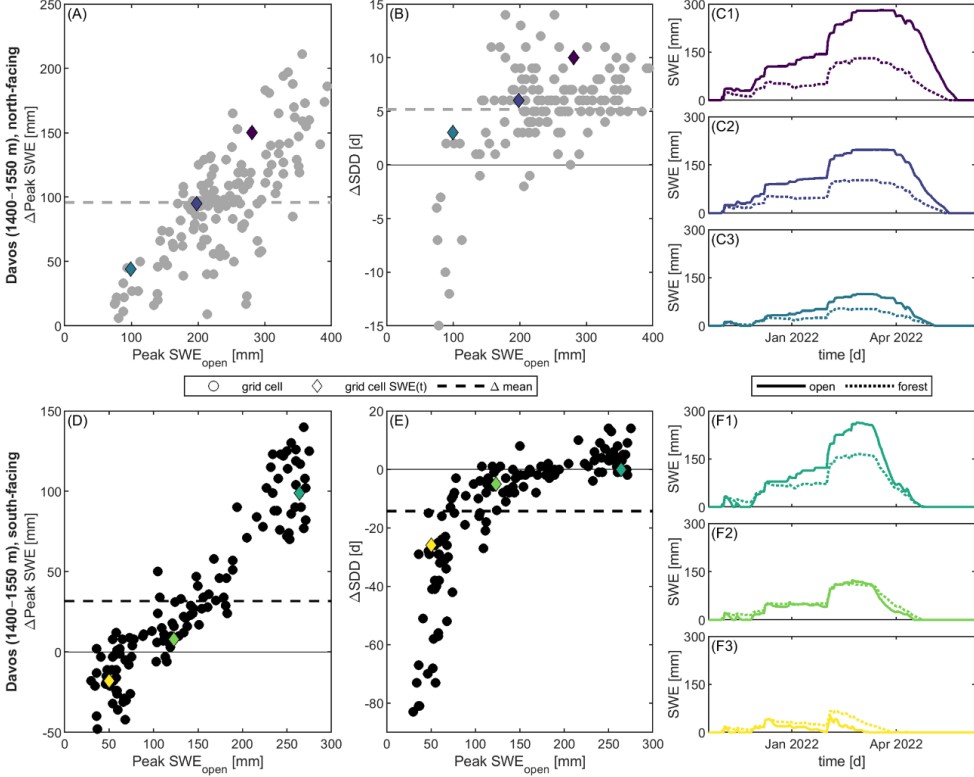


**Figure 10: An evaluation of the contrasting conditions in terms of ΔPeak SWE and ΔSDD relative to values of peak $SWE_{open}$ for all grid cells of the 1400–1550m elevation band on north-facing (A-B) and south-facing (D-E) aspects. Additionally, SWE time series of forested and open fractions are presented for three selected grid cells on north-facing aspects (C1-C3) and south-facing aspects (F1-F3), respectively.**

On south-facing slopes, the overall (average) effect of forest amounts to a positive ΔPeak SWE (Fig. 10D) and a negative ΔSDD (Fig. 10E) in line with results shown in Fig. 6F and Fig. 7F. However, grid cells that contribute to the negative ΔSDD are predominantly those that feature a low peak $SWE_{open}$ (Fig. 10, F2-F3). In contrast, grid



cells that contribute to the positive $\Delta$Peak SWE are predominantly those that feature a high peak SWE$_{open}$ (Fig. 10, F1). So, despite the overall effect, there may not be a single grid cell that features both a positive $\Delta$Peak SWE and
a negative $\Delta$SDD at the same location. On north-facing slopes, there is less potential for pronounced ablation during accumulation, which is why grid cells with peak SWE$_{open}$ values below 100 mm are largely missing. This promotes higher $\Delta$Peak SWE (Fig. 10A) and essentially inhibits negative $\Delta$SDD values (Figs. 10B, C1-C3), resulting in an overall much higher $\Delta$Peak SWE and a positive $\Delta$SDD (cf. Fig. 6F and Fig. 7F).

These considerations can also be extrapolated to explain $\Delta$SDD differences between years and regions. For
instance, HY2020 was a snow-scarce year with low accumulation (i.e., peak SWE$_{open}$), particularly in the Interlaken and Schwyz regions. This resulted in many grid cells with very negative $\Delta$SDD values, even on north-facing slopes, leading to a reduced aspect dependence in both regions (Fig. 7B-C). On the other hand, snow-rich and/or cold years that lead to pronounced $\Delta$Peak SWE (Fig. 9A, 2018/2019/2021; Davos region) provide less potential for many grid cells with very negative $\Delta$SDD values, hence showing overall neutral or slightly positive
$\Delta$SDD even on south-facing slopes (cf. Fig. 9B). Finally, in the Jura region, conditions are generally different, with, on average, half the amount of peak SWE compared to all the other regions. This favors conditions similar to those in panels F2 and F3 of Fig. 10, with negative $\Delta$SDD irrespective of elevation and aspect. In addition, the topography of the Jura Mountains differs from that of the Alps as the highest elevations are plateau-like, i.e., relatively flat.

The combined effects of forest cover, topography, and interannual weather variability on forest snow dynamics, as discussed above, provide further context for past observations by Koutantou et al. (2022), who found significant differences in small-scale snow height distributions between two opposing forested north- and south-facing slopes within the study area. Subsequently, Mazzotti et al. (2023) were able to link these differences to ablation from early-season insolation on south-facing slopes in the absence of topographic shading. Our results demonstrate that
these processes, initially investigated at the scale of individual trees, also affect snow distribution dynamics at much coarser spatial scales and over large extents. The relevance of both forest cover and topography on snow cover dynamics beyond the valley scale has also been noted by Broxton et al. (2020), who found SWE differences to increase between areas with more or less forest cover as winter progressed, influenced by terrain shading or the lack thereof on opposing slopes.

**4.2    Forest structure effects on snow storage**

Distinct aspect- and elevation-dependent patterns of $\Delta$Peak SWE are apparent in Fig. 6 and Fig. 9A. Not surprisingly, these patterns generally match those of leaf area index (LAI, cf. Fig. A3), confirming that canopy snow processes have a considerable impact on snow accumulation on the ground. For example, in the Rhone, Gotthard, and Davos regions, LAI is consistently higher on north-facing aspects within the 1100–1700 m range,
correlating with increased $\Delta$Peak SWE (Fig. 6), regardless of how snow-rich a winter is (Fig. 9A). In the Interlaken and Schwyz regions, there is little aspect dependency of LAI, which explains the absence of corresponding $\Delta$Peak SWE patterns (B-C, Fig. 6). Above 1700m across the study domain, LAI rapidly decreases, explaining why, even though cumulative annual snowfall further increases with elevation, differential peak SWE storage does not. These interrelations demonstrate the role of snow interception, which is known to scale with LAI (Hedstrom and
Pomeroy, 1998). The longer and more snow is trapped in the canopy, the more intercepted snow will sublimate back into the atmosphere instead of contributing to snow accumulation on the ground (Lundquist et al., 2021; Pomeroy et al., 1998). Hence, ablation processes aside, peak SWE should, in first order, decrease with increasing LAI, explaining the correlation between the patterns in Fig. 5 and Fig. A3. However, ablation can make a difference. Areas above 1250 m in the Jura region provide an example where peak SWE is persistently higher
inside forests (Fig. 6A) due to considerably more snowmelt during the accumulation period outside of forests, equivalent to the examples shown in Fig. 10, F2-F3. This phenomenon has also been reported during winter storms with rain on snow, where ablation in forests was limited because of lower wind speeds that reduced the magnitude of turbulent exchanges at the snow surface (Marks et al., 1998). Alternatively, slightly higher SWE accumulation in forests could also result from differences in sublimation (Gelfan et al., 2004).

The current forest structures across the Alps resulted from past land-use-driven human interferences, as well as natural disturbances such as bark beetles, wildfires, and windthrow (Bebi et al., 2017). Coniferous forests are prevalent, and this biome is where future changes in forest structure are likely to be most pronounced (Seidl et al., 2017), in turn affecting forest snow dynamics. For example, beetle infestations can reduce LAI and thus interception, leading to increased accumulation on the ground (Boon, 2012). In addition, forests are unevenly
altered by wildfires, creating patches of varying burn severity (Koshkin et al., 2022), where an increase in overstory mortality is associated with increased sub-canopy snow accumulation (Maxwell and St Clair, 2019). Hence, especially in forests with dense canopies, local disturbances could lead to an overall increase in peak SWE storage. In the study domain, this applies specifically to north-facing slopes in the central Alps (Rhone, Gotthard, and Davos regions, cf. Fig. A3). However, topography also influences how forest disturbances affect snow cover, as
both terrain and canopy affect the amount of sunlight reaching the snowpack (Rinehart et al., 2008). On north-facing slopes, sparser forests often increase snow storage due to limited winter insolation and thus ablation



(Broxton et al., 2020; Harpold et al., 2020). In contrast, south-facing slopes receive more sunlight, making the effects of disturbance more complex. Denser forests can reduce snow storage via interception and increased longwave radiation (Broxton et al., 2020), while canopy removal may also accelerate ablation and reduce peak SWE (Mazzotti et al., 2023). With forest disturbances projected to increase (Seidl et al., 2017), understanding their effects on snow storage in complex terrain is crucial, though beyond this study's scope. Currently, most of such research originates from North America, driven by concerns related to water and fire management (e.g., Dickerson-Lange et al., 2023, 2021; Lewis et al., 2023; Moeser et al., 2020; Pomeroy et al., 2012). However, given the sensitivity of snow process interactions to forest structure, climate, and topography, corresponding studies from other regions with different environmental conditions are equally vital.

### 4.3 The hydrological relevance of forest snow

Forest snow can account for more than 30 % of the total snow storage, yet surprisingly minor differences result when comparing simulations with and without forest. With no forest cover at all, total peak snow storage would only increase by a few percent (Fig. 4). This may seem surprising, considering in how many ways forests influence snow processes. However, while approximately 40 % of the study area is forest-covered or affected by its presence, more than 50 % of this is located at elevations below 1000 m. This considerably constrains the effect of forests on snow storage at the national level in absolute quantities, since most of the total mass is accumulated higher up in the mountains. Secondly, while peak SWE is increased due to a lack of canopy interception (Fig. 6), pronounced ablation in the open during accumulation on south-facing slopes might offset some of this increase (cf. peak SWE$_{open}$ in Figs. 10A and 10D; Sect. 4.1). Additionally, different meteorological conditions in the open (e.g., higher wind speeds) will result in increased sublimation losses from the ground (Strasser et al., 2008). Combined, these factors may explain why only a relatively small increase in total peak SWE storage would result if there were no forests in the study domain.

Even if the effect on total snow storage is relatively small, the presence of forest implies considerable changes to the spatial distribution and temporal evolution of snow cover, as is evidenced by the complex patterns seen in Figs. 6, 7, 8, and 9, and as discussed above. The increased spatial heterogeneity of forest snow cover is known to disperse the occurrence of snowmelt runoff over time. As a result, downstream rivers can be expected to have a flatter seasonal hydrograph, reducing peak flow (Pomeroy et al., 2012) and decreasing the risk of spring snowmelt flooding (Hendrick et al., 1971). Recent modeling studies have shown that at the watershed scale in the US, increasing forest heterogeneity leads to increased low stream flows into late summer (Sun et al., 2018). The extent of these runoff changes depends on local climatic conditions and forest structure (Currier et al., 2022).

Our findings confirm spatially heterogeneous effects of forest cover on the timing of snowmelt runoff, which shifts depending on aspect and elevation (Fig. 8). Interestingly, findings in the central Alps (D-F, Fig. 8) are in line with Ellis et al., (2011) who found that forests substantially delayed snowmelt onset on south-facing slopes, whereas on north-facing slopes, snowmelt onset occurred at the same time but progressed much more quickly in forests; this in contrast to other regions where aspect dependent responses were largely missing. Not surprisingly, patterns of ΔCT (Fig. 8) corresponded to those of ΔSDD (Fig. 7), indicating that dependencies discussed in Sect. 4.1 also have larger-scale impacts on snowmelt runoff generation. Still, other factors, such as soil characteristics, also affect how heterogeneous snow melt runoff in montane forests translates into a hydrograph response (Pomeroy et al., 2012; Redding and Devito, 2011).

### 4.4 Opportunities and limitations of a simulation-based analysis

Using model output data in this study was a necessity, as equivalent measurements do not exist. Even in areas that undergo periodic ALS acquisitions (Dwivedi et al., 2024; Painter et al., 2016), temporal coverage would likely not be sufficient to reveal the level of detail discussed here (e.g., Sect. 4.1). The approach of using data from physically based forest snow models to analyze effects that are difficult to observe is not new. Exploiting hyper-resolution simulations with FSM2 (2 m), Mazzotti et al. (2023) advanced the understanding of forest snow process interactions in complex terrain at the level of individual mass and energy balance components, providing the basis for this work. Other recent studies used similar approaches, for example, to highlight the relevance of fine-scale forest structure, which governs a range of relevant forest snow process interactions (Broxton et al., 2021). Even 15 years ago, modeling studies were already instrumental in forming today's understanding of how snow dynamics are affected by the presence of forest cover in mountainous topography (e.g., Pomeroy et al., 2012; Strasser et al., 2011).

Of course, the FSM2oshd model used here, like any model, does not provide perfectly accurate simulations of the actual conditions; however, the model has previously been shown to accurately represent forest snow processes across various sites, years, climates, and spatial resolutions (Mazzotti et al., 2020a, 2020b, 2023). In addition, data assimilation strategies ensured that the model remained consistent with daily snow depth observations from ~500 stations (Cluzet et al., 2024; Mott et al., 2023). Further validation of FSM2oshd simulations in this study aimed to provide extra confidence in the approaches used for upscaling the physical representations of meter-scale forest snow processes to 250 m resolution (cf., Mazzotti et al., 2021). With no ALS data available, PlanetScope satellite



imagery proved an effective solution for validating FSM2oshd over such a large area at the necessary spatial resolution. This demonstrated the ability of FSM2oshd to capture complex patterns of spatiotemporal snow cover dynamics (Figs. 3, A1, A2), expressed by a persistently low MAE (Table A1), which also attests to the capabilities of the methodology used for ingesting three-dimensional forest structure data (Mott et al., 2023; Webster et al., 2023).

While FSM2oshd can represent lateral redistribution processes (Quéno et al., 2024), these process representations are only effective at higher spatial resolutions and would require a grid spacing of 100 m or higher. Here, we prioritized a large spatial extent over a high resolution needed to include redistribution processes. Including snow redistribution processes might have provided additional insights, even though subalpine forests in the Alps are typically surrounded by narrow, steep terrain, which provides shelter from strong winds. Nevertheless, we acknowledge that the model setup used here may underpredict sublimation losses of snow in open terrain, particularly in wind-exposed settings at high elevation, i.e., above the tree line.

## 5    Conclusion

This work provided a comprehensive analysis of how mountain forests affect snow cover dynamics across a large 58'000 km$^2$ area with variable climate and complex topography in the central European Alps. Data for this study were obtained from simulations using the process-based FSM2oshd forest snow (Mazzotti et al., 2020b, a; Mott et al., 2023). FSM2oshd has undergone extensive validation, both here and in several preceding studies, demonstrating its capability to simulate snow cover dynamics in complex mountainous landscapes accurately. Simulations were conducted at an hourly temporal and 250 m spatial resolution over eight years. The study area extended from the lowlands north of the Alps to the lowlands south of them, encompassing elevations between 184 m and 4,806 m. Meteorological forcings were available from the high-resolution numerical weather prediction system COSMO, run by MeteoSwiss, which was debiased and downscaled using the approaches detailed in Mott et al. (2023). Canopy data were derived from high-resolution lidar acquisitions, enabling the detailed representation of radiation transmission through the three-dimensional canopy structure using CanRad (Webster et al., 2023).

Our analysis revealed extensive spatial variability of snow cover dynamics in response to variable forest cover, topography, and climatological conditions (Figs. 6 and 7). A particularly striking finding is how differently snow in north- and south-facing terrain is affected by the presence of forest cover. Effects that were previously only known from meter-resolution observations or simulations, particularly the decisive role of early-season melt (Mazzotti et al., 2023), were carried over to the coarser-scale simulations analyzed here, with considerable consequences for runoff generation from snowmelt at catchment scales (Figs. 8 and 10). Variability between years was found to be strong enough to shift or even reverse typical trends with elevation, aspect, between regions, or both, where low-snow years accentuated relative differences in the effects of forest on snow cover (Fig. 9). These results highlight that extensive datasets across large spatial and temporal extents, be it model-based or observational, are necessary to fully embrace in which intricate ways key factors interact to control forest snow processes. Given the hydrological and ecological relevance of forest snow in a changing environment, our work encourages ongoing efforts to advance modelling and observational approaches applicable at these large spatiotemporal scales.



*Code availability:* Code for the FSM2oshd snow model is available at https://github.com/oshd-slf/FSM2oshd.
COSMO forcing data is available from MeteoSwiss at https://www.meteoswiss.admin.ch/services-and-publications/service/weather-and-climate-products/weather-forecasts.html

*Data availability:* FSM2oshd model output data used in analyses of this manuscript will be made available upon request. PlanetScope satellite imagery is available at https://www.planet.com/products/satellite-monitoring/

*CRediT authorship contribution statement*: **Vincent Haagmans:** Conceptualization, Data Curation, Formal Analysis, Investigation, Methodology, Software, Validation, Visualization, Writing – Original Draft Preparation, Writing – Review & Editing. **Giulia Mazzotti:** Conceptualization, Methodology, Software, Supervision, Writing – Review & Editing. **Clare Webster:** Methodology, Software, Writing – Review & Editing. **Tobias Jonas:** Conceptualization, Funding Acquisition, Investigation, Methodology, Project Administration, Resources, Software, Supervision, Validation, Writing – Original Draft Preparation, Writing – Review & Editing.

*Competing interests:* The authors declare that they have no conflict of interest.

*Acknowledgements:* We thank SLF's Operational Snow Hydrological Service for providing the simulations that served as the basis for this study. GM received funding from the Swiss National Science Foundation, grants 500PN_202741 and 5R5PN_225378. Generative AI tools, including Grammarly, ChatGPT, were used to assist in improving the clarity, grammar, and language of this manuscript. These tools were not used to generate original scientific content (e.g., figures), perform data analysis, or draw conclusions. All intellectual content, interpretations, and final decisions reflect the authors' work.

Appendix

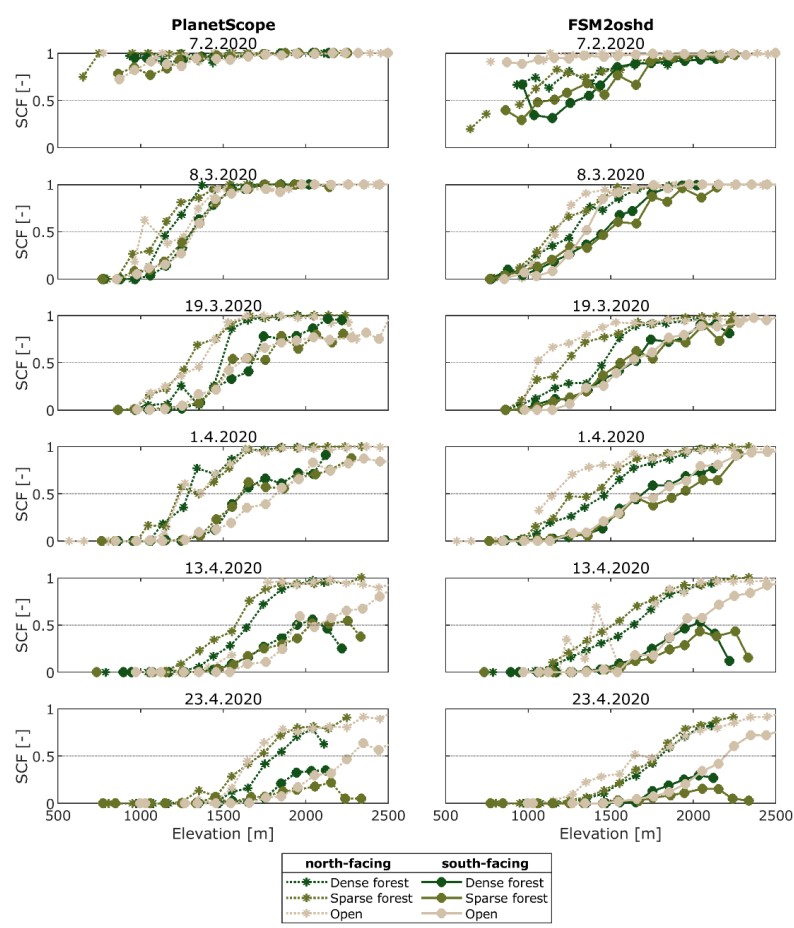

**Figure A1: Comparison of observed (PlanetScope) and modeled (FSM2oshd) snow cover fraction (SCF) for the Davos focus region during winter and spring of 2020. SCF profiles are shown separately for three land cover types (i.e., dense forest, sparse forest, and open area) on north- or south-facing slopes.**



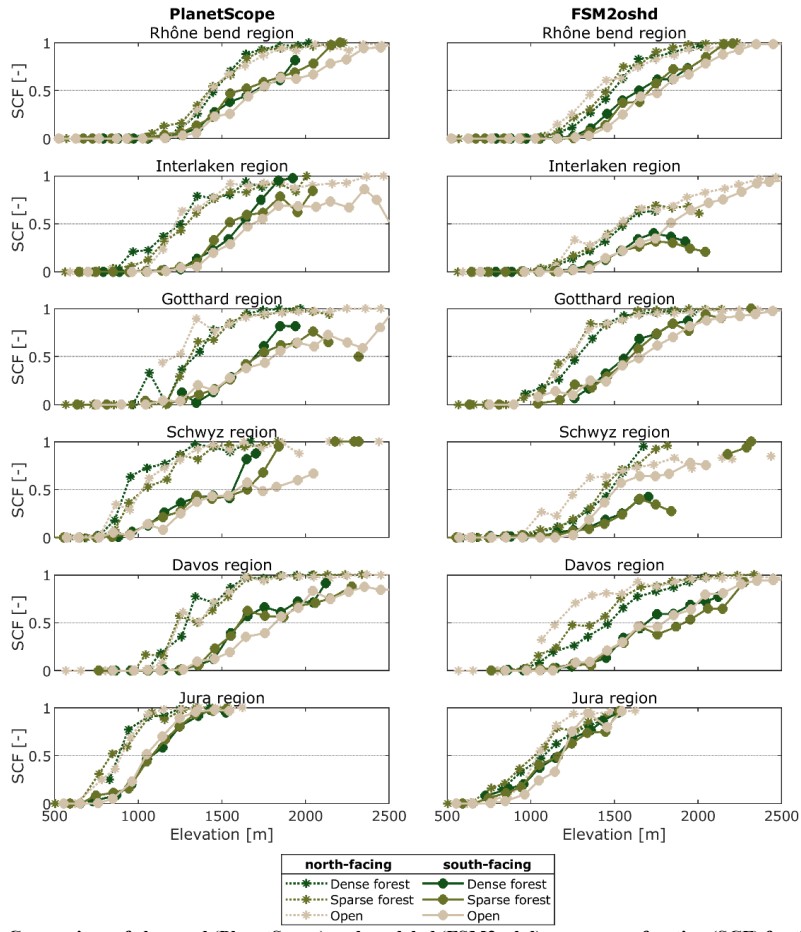

**Figure A2: Comparison of observed (PlanetScope) and modeled (FSM2oshd) snow cover fraction (SCF) for the six focus regions at the end of winter (approx. April 1st). SCF profiles are shown separately for three land cover types (i.e., dense forest, sparse forest, and open area) on north- or south-facing slopes.**





**Table A1: Performance of the FSM2oshd model to simulate snow cover fraction relative to observations expressed by the mean absolute error (MAE). The percentage of grid cells in an acquisition, assessed by land cover type, is also indicated.**

| Focus region | Date | Open area MAE (% grid cells) | Sparse forest MAE (% grid cells) | Dense forest MAE (% grid cells) |
|---|---|---|---|---|
| Rhone bend | 2020-04-01 | 0.158 (4.8 %) | 0.125 (9.4 %) | 0.128 (16.5 %) |
| Interlaken region | 2020-04-01 | 0.216 (5.0 %) | 0.229 (10.4 %) | 0.219 (24.5 %) |
| Gotthard region | 2020-04-03 | 0.169 (3.7 %) | 0.152 (11.8 %) | 0.181 (22.1 %) |
| Schwyz region | 2020-04-01 | 0.255 (4.7 %) | 0.269 (7.2 %) | 0.399 (13.0 %) |
| Jura region | 2022-02-10 | 0.207 (4.9 %) | 0.142 (9.6 %) | 0.200 (9.5 %) |
| Davos region | 2019-03-29 | 0.165 (3.2 %) | 0.118 (14.9 %) | 0.132 (14.4 %) |
| Davos region | 2020-02-07 | 0.046 (1.6 %) | 0.196 (6.2 %) | 0.171 (6.8 %) |
| Davos region | 2020-03-08 | 0.096 (2.7 %) | 0.138 (12.4 %) | 0.171 (12.6 %) |
| Davos region | 2020-03-19 | 0.170 (3.0 %) | 0.110 (12.1 %) | 0.127 (12.9 %) |
| Davos region | 2020-04-01 | 0.136 (3.9 %) | 0.102 (11.8 %) | 0.138 (12.6 %) |
| Davos region | 2020-04-13 | 0.131 (3.9 %) | 0.113 (11.3 %) | 0.120 (14.5 %) |
| Davos region | 2020-04-23 | 0.138 (4.8 %) | 0.126 (11.6 %) | 0.134 (12.8 %) |
| Davos region | 2021-04-01 | 0.168 (6.1 %) | 0.122 (20.4 %) | 0.153 (22.4 %) |
| Davos region | 2022-03-28 | 0.168 (4.2 %) | 0.122 (15.6 %) | 0.141 (17.7 %) |
| Davos region | 2023-04-06 | 0.167 (5.2 %) | 0.152 (19.0 %) | 0.169 (19.7 %) |
| Davos region | 2024-04-06 | 0.171 (4.7 %) | 0.117 (16.9 %) | 0.132 (14.9 %) |



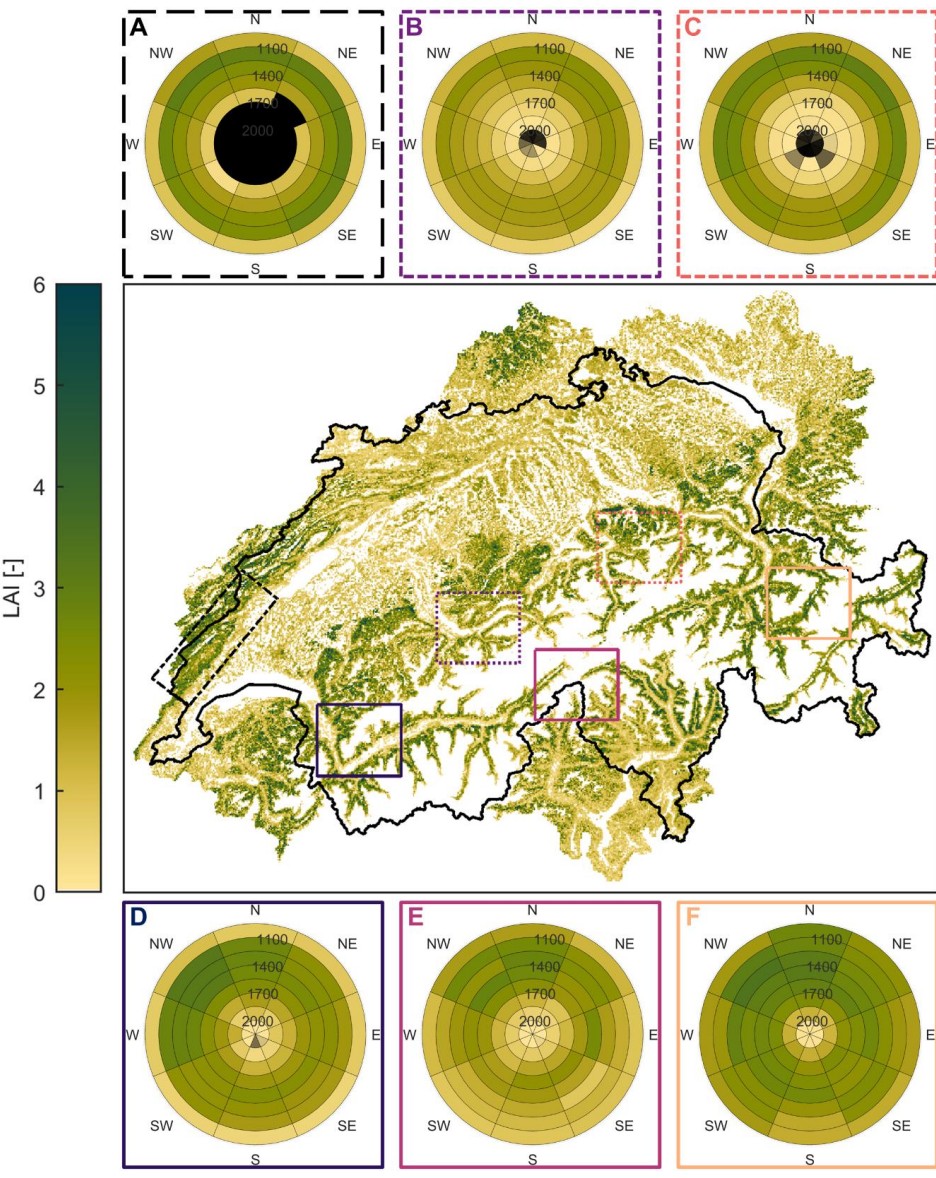

**Figure A3: Leaf area index (LAI). For each of the six focus regions, a polar plot is provided (A–F). These show the average ΔSDD for eight 150 m elevation bands (950–2150 m) across eight aspect classes, where the labels refer to the directions in which slopes face.**



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
