# Peer review of "How montane forests shape snow cover dynamics across the central European Alps"

_EGUsphere, 2025_

## Referee Comment (RC1)

**Review of Haagmans et al., https://doi.org/10.5194/egusphere-2025-3843**

Based on a modelling study of subcanopy and open-terrain snow cover for 8 years in a large Central Alpine domain, the authors propose a detailed analysis of the influence of different factors on the contrasted snowpack dynamics between forested and open areas. The factors analysed encompass aspect, altitude, location and interannual climate variability.

The study is well written, very well illustrated, and well structured. The impacts of forests on subcanopy snow have to my knowledge never been assessed with such level of analysis and detail over such a large spatial scale, making them very relevant for publication and of high interest for hydrological applications.

I have only **minor comments** which in my opinion should be taken into account prior to publications.

**Main minor comments:**

- In section 2.2, a description of how SCF is derived in the OSHD simulations is missing, making it hard to fully understand and assess the relevance of the evaluation carried out in Section 3.1
- The approach and results are not sufficiently discussed with respect to a previous publication that imo contributed to prepare the grounds for the present study and drew relevant conclusions for large spatial scales, namely Lundquist et al., 2013 (whom the authors cite). Reference and an assessment of difference/progress beyond this work should be made in the Discussion.

**Specific comments:**

L 259: it should de 13th April and not 14th April

L 364-366 : « *In contrast, forests on north-facing slopes advanced CT by up to two weeks, which also applied more generally above 1850 m in all regions, regardless of aspect.* » Could the fact that forest is likely sparser above 1850 m, play a role in explaining this? If relevant, the effect of canopy density could be a bit more discussed with respect to this result.

L 400-403: « Here, it is likely the difference in accumulation that drives the overall effect of forests on snow persistence ». I think the affirmation is a bit stronger than what the observation tells, and maybe a reformulation could be appropriate, like « Here, the difference in accumulation likely has an important contribution to the overall effect of forests on snow persistence ».

L 458: I think Fig 6 is meant instead of Fig 5

L 525 : « Even 15 years ago, modeling studies were already instrumental in forming today's understanding of how snow dynamics are affected by the presence of forest cover in mountainous topography » The formulation is weird and maybe a reformulation should be attempted.

L 533 : suggestion to replace « extra » by additional

---

## Author Comment (AC1)

We thank Reviewer 1 for their positive and constructive feedback, which helped improve our manuscript. All their comments will be addressed according to this reply. Reviewer comments are in *blue cursive text*. Changes or updates to the manuscript as a response to a comment are in **bold text.**

Main minor comments

*- In section 2.2, a description of how SCF is derived in the OSHD simulations is missing, making it hard to fully understand and assess the relevance of the evaluation carried out in Section 3.1*

Thank you for pointing this out. We have added the following sentences at the end of Section 2.2 to address this comment. Further details on snow cover fraction parameterizations are outlined in Mott et al. (2023).

**"For open areas, the snow cover fraction (SCF) is defined using the algorithm by Helbig et al. (2021). The SCF of forested areas is parameterized from snow depth using a hyperbolic tangent model (Essery, 2015)."**

*- The approach and results are not sufficiently discussed with respect to a previous publication that imo contributed to prepare the grounds for the present study and drew relevant conclusions for large spatial scales, namely Lundquist et al., 2013 (whom the authors cite). Reference and an assessment of difference/progress beyond this work should be made in the discussion.*

To address this comment, we revise L438-440:

**"Subsequently, Mazzotti et al. (2023) were able to link these differences to ablation from early-season insolation on south-facing slopes in the absence of topographic shading. By considering additional topography-driven processes and including detailed canopy structure information, both aforementioned studies advanced the conceptual framework of Lundquist et al. (2013), who focused on the comparison of forested vs. non-forested conditions at the site scale. Here, our results demonstrate that the combined effects of these processes, initially investigated at the scale of individual trees, also affect snow distribution dynamics at much coarser spatial scales and over large extents."**

Specific comments

*L 259 : it should de 13th April and not 14th April*

Good catch. Corrected to "**April 13th**"

*L 364-366 : « In contrast, forests on north-facing slopes advanced CT by up to two weeks, which also applied more generally above 1850 m in all regions, regardless of aspect. Could the fact that forest is likely sparser above 1850 m, play a role in explaining this? If relevant, the effect of canopy density could be a bit more discussed with respect to this result.*

We find no clear correlation between sparser forests, here expressed as a f(LAI), and differences in CT, neither across the whole domain nor for the five focus regions individually; i.e., negative or positive CT occur approx. equally at low LAI < 2:

[Figure]

*L 400-403 : « Here, it is likely the difference in accumulation that drives the overall effect of forests on snow persistence ». I think the affirmation is a bit stronger than what the observation tells, and maybe a reformulation could be appropriate, like « Here, the difference in accumulation likely has an important contribution to the overall effect of forests on snow persistence ».*

Given that in north-facing slopes the melt rates in the open generally exceed those in the forests (c.f. Fig. 10, C1-C3), it has to be the difference in accumulation that drives the overall effect (which is snow persists longer in the open). We therefore think that our statement is adequate.

*L 458 : I think Fig 6 is meant instead of Fig 5*

Indeed. Corrected to "**Fig. 6**".

*L 525 : « Even 15 years ago, modeling studies were already instrumental in forming today's understanding of how snow dynamics are affected by the presence of forest cover in mountainous topography » The formulation is weird and maybe a reformulation should be attempted.*

We reformulated this sentence addressing corresponding comments of both reviewers 1 and 2:

**Throughout the past decades, modeling studies have played an important role in shaping today's understanding of both specific forest snow processes (e.g., Pomeroy et al., 1998) and how process interactions in montane forests modulate the subcanopy snow cover (e.g., Pomeroy et al., 2012; Strasser et al., 2011).**

*L 533 : suggestion to replace « extra » by additional*

Changed as suggested

**References**

Essery, R.: A factorial snowpack model (FSM 1.0), Geosci. Model Dev., 8, 3867–3876, https://doi.org/10.5194/gmd-8-3867-2015, 2015.

Helbig, N., Schirmer, M., Magnusson, J., Mäder, F., van Herwijnen, A., Quéno, L., Bühler, Y., Deems, J. S., and Gascoin, S.: A seasonal algorithm of the snow-covered area fraction for mountainous terrain, The Cryosphere, 15, 4607–4624, https://doi.org/10.5194/tc-15-4607-2021, 2021.

Lundquist, J. D., Dickerson-Lange, S. E., Lutz, J. A., and Cristea, N. C.: Lower forest density enhances snow retention in regions with warmer winters: A global framework developed from plot-scale observations and modeling, Water Resour. Res., 49, 6356–6370, https://doi.org/10.1002/wrcr.20504, 2013.

Mazzotti, G., Webster, C., Quéno, L., Cluzet, B., and Jonas, T.: Canopy structure, topography, and weather are equally important drivers of small-scale snow cover dynamics in sub-alpine forests, Hydrol. Earth Syst. Sci., 27, 2099–2121, https://doi.org/10.5194/hess-27-2099-2023, 2023.

Pomeroy, J., Fang, X., and Ellis, C.: Sensitivity of snowmelt hydrology in Marmot Creek, Alberta, to forest cover disturbance, Hydrol. Process., 26, 1891–1904, https://doi.org/10.1002/hyp.9248, 2012.

Pomeroy, J. W., Parviainen, J., Hedstrom, N., and Gray, D. M.: Coupled modelling of forest snow interception and sublimation, Hydrol. Process., 12, 2317–2337, https://doi.org/10.1002/(SICI)1099-1085(199812)12:15%253C2317::AID-HYP799%253E3.0.CO;2-X, 1998.

Strasser, U., Warscher, M., and Liston, G. E.: Modeling Snow–Canopy Processes on an Idealized Mountain, J. Hydrometeorol., 12, 663–677, https://doi.org/10.1175/2011JHM1344.1, 2011.

---

## Author Comment (AC2)

We thank Reviewer 2 for their positive and constructive feedback, which helped improve our manuscript. All their comments will be addressed according to this reply. Reviewer comments are in *blue cursive text*. Changes or updates to the manuscript as a response to a comment are in **bold text.**

*L13: is equally right word here? The controls would have different dominance in different regions*

This statement was based on Mazzotti et al. (2023); however, we have removed the word "**equally**" to address your concern.

*Table 1: where does the data for variables in table one originate from?*

The data in Table 1 are computed based on the topographic, land cover data, and meteorological driving data used as input for FSM2oshd. The datasets are described in more detail in Chapter 2.2 as well as in Mott et al. (2023). The table caption was revised as:

"**Topographical, land cover, and meteorological characteristics of the six focus regions as outlined in Figure 1. Meteorological characteristics are based on the model forcings evaluated over the study period and grid cells with forest cover. Topographical characteristics are taken from land cover datasets used for our simulations; See Chapter 2.2 for further information.**"

*L166: excluding images with intercepted snow is not explained. what is the rule for saying there is interception? Would this not create a bias for the analysis, as I'd expect interception to be often present in the mid-winter?*

Thanks for the comment. In the majority of cases, intercepted snow only survives a few days in the canopy before either unloading/dripping or sublimating, i.e., most of the time there is no snow in the canopy, particularly mid-February onwards. So, these conditions in Switzerland are not a major constraint. The reasoning for not using PlanetScope imagery affected by intercepted snow is that we wanted to exclude possible misclassification of snow in the canopy for snow on the ground. Determining whether or not snow was present in the canopy, on the other hand, was fairly obvious.

To address your comment, we added the above reasoning to condition ii), which now reads:

"**ii) no intercepted snow was visible in the canopy to avoid possible misclassification of snow in the canopy for snow on the ground**".

*L202: 30 000 grid cells is a lot to verify manually! would you expect any biases there, and can you comment if the process is reproducible*

Certain steps in the procedure were specifically aimed at maximizing reproducibility of the manual assessment. In particular, working with seven intuitive classes (step 3) and doing separate evaluations according to land cover type (step 2) made the attribution relatively easy, even if edge cases occurred at times. Additionally, when assessing a PlanetScope composite, we actively ensured that we covered all aspects and elevations across the area, which would have otherwise potentially led to biases due to over- or undersampling in a particular elevation band or aspect. We are thus fairly confident that these results could be reproduced with a very small error margin, even if the assessment was redone from scratch.

*L250-253: the interpretation here having statements like typically and generally is somewhat vague and difficult to verify from figures. Can you give more concrete examples to show the point you are making, as in earlier part of the paragraph.*

We agree that we could have been more concrete in our presentation. To address this comment, we have revised L250-253 as follows:

"**Remarkably, differences in SCF profiles between the three land cover types (open area, sparse forest, and dense forest) remained typically within ~50 m, with some notable exceptions between forested and open areas in 2022 and 2023. All the above findings from observations were generally well represented by the FSM2oshd model, except that in some cases the model overestimated the differences between SCF profiles of the three land cover classes, particularly on the north-facing slopes in 2020 and 2024.**"

*L259: 13th instead of 14th?*

Indeed. Corrected to "**April 13th**"

*L263: Perhaps "showed similar patterns" is an overstatement here. Id argue that patterns for sites 2, 4 and 6 are quite different for observed vs simulated.*

While we agree that data in Fig A2 shows some model limitations (which we cover in line 267ff), this first overview statement was intended to address the observed SCF only (left side of Fig. A2). To mitigate such a possible misunderstanding, we reformulated the onset of the paragraph to now read:

"**The observation-based SCF profiles (left side of Fig. A2) showed similar patterns across the six focus regions, and differences were generally minor compared to the variability between aspects and years (cf. Fig. 3), the Jura region, however, being an exception. Furthermore, differences of SCF profiles between aspects were notably smaller in the Rhone bend and Jura regions compared to the other regions.**"

*L289-306: what area do you use to calculate the SWE: the total study area, or the snow covered area for a given time? This will greatly influence the SWE numbers. Furthermore, I'm used to SWE being reported in units of [mm], corresponding kg/m2, i.e. mass per area. Where you talk about SWE in units of mm and km3 interchangeably, I recommend bringing consistency in SWE units throughout the manuscript.*

We totally agree, following the suggestion, we have revised Figure 4 as shown below:

[Figure]

The associated text in Chapter 3.2 was revised as follows:

**"The maximum amount of water stored as snow across the study area during a season averaged at 126 mm and ranged from 87 mm to 192 mm (Fig. 4A), corresponding to a total volume of water between 4.8 km³ and 10.6 km³. Forested areas had an average peak SWE storage of 56 mm, which during the study period ranged from 28 mm to 108 mm, corresponding to a total volume of water between 0.7 km³ and 2.6 km³ (Fig. 4B)."**

and:

**"If forest cover was unaccounted for over the entire model domain, total SWE storage would, at its peak, increase by only 5 mm on average, which is 3.7% of the current mean total peak SWE (cf. Fig. 4A and Fig. 4D). The increase of the maximum SWE storage, in this scenario, reached 9 mm during HY2021 and occurred 11 days later than the total peak SWE during that year (Fig. 4A)."**

*Figure 6: recommend to add letters also to the map next to boxes: dashes and box colors not very obvious indicator*

Thank you for this suggestion, which enhances the readability of the figures.

[Figure]

We implemented this for all similar **figures (7, 8, A3)**.

*L403-434: figure 10 and associated interpretation is difficult to follow. Even after repeated attempts, I don't fully understand the logic of the figure, and how it supports the analysis. This might be only me, but think about any simplifications or supporting information that would make this easier to digest. Or is this necessary to include in the first place, as you already have a lot of analysis and results to talk about. Maybe removing altogether would not compromise the main message, but would make the paper more short and concise?*

We acknowledge that Figure 10 and its associated interpretation may be challenging to follow. However, we deem it an essential part of the discussion, as it demonstrates how overall effects are formed by the cell-to-cell variability, which arises due to local differences in topography and forest structure. As an example, on south-facing slopes the overall average effect of forest amounts to a positive ΔPeak SWE and a negative ΔSDD, while there may not be a single grid cell that actually features both at the same time. Such results constitute a key asset of our study as they reveal, maybe for the first time, that understanding forest snow effects does require accounting for small scale variability of the involved processes even at scales far beyond typical experimental studies. Nevertheless, we acknowledge that we may need to offer better context when introducing the figure and have hence extended the paragraph starting in line 403:

**"However, spatiotemporal differences in accumulation and ablation, as well as their variations between forested and open terrain, are more intricate than the above description of the overall effect suggests. In Figure 10, we show that conditions averaged over large scales are driven by the range of grid-cell-to-grid-cell variability, rather than representing conditions prevailing in any majority of grid cells. To this end, we present an evaluation of the contrasting conditions between south and north-facing grid cells from only one region (Davos) and one elevation band (1400–1550 m). There is considerable variability in the amount of snow that builds up during the accumulation period in the open (peak SWE$_{open}$), which is primarily due to local differences in concurrent melt, modulated by terrain shading and variable microclimatic conditions. Spatial variability in snowfall is less influential in this regional evaluation (data not shown)."**

the caption of Figure 10:

**"SWE time series of forested and open fractions are presented for three selected grid cells on north-facing aspects (C1-C3) and south-facing aspects (F1-F3), see colored marker in A, B, D, E, and corresponding line plots in C1-C3 and F1-F3."**

and added some additional pointers in the description of Figure 10:

**"On south-facing slopes, the overall (average) effect of forest amounts to a positive ΔPeak SWE (Fig. 10D, dashed line) and a negative ΔSDD (Fig. 10E, dashed line) in line with results shown in Fig. 6F and Fig. 7F. However, grid cells that contribute to the negative ΔSDD are predominantly those that feature a low peak SWE$_{open}$ (Figs. 10E, F2-F3). In contrast, grid cells that contribute to the positive ΔPeak SWE are predominantly those that feature a high peak SWE$_{open}$ (Figs. 10D, F1)."**

*L487-498: One process to discuss further here is how snow unloading from the canopy is simulated, and are the unloading routines well validated? Fast unloading means less time for interception sublimation, and less difference between open and forested snow.*

Yes, absolutely. In FSM2oshd, canopy snow processes are based on parameterizations going back to Hedstrom and Pomeroy (1998). These were later adapted for use with local canopy structure parameters following Mazzotti et al. (2020b) . The updated canopy snow routines have been carefully evaluated and shown to simulate meter-scale snow depth distributions that closely match observations (Mazzotti et al., 2020b, a).

To address your comment, we added the following remark:

**"This considerably constrains the effect of forests on snow storage at the national level in absolute quantities in two ways: at those elevations unloading of intercepted snow happens usually within a few days leaving little time for respective sublimations losses. Moreover, most of the total mass is accumulated higher up in the mountains, leaving differences between forested and open at lower elevations only a small share."**

*L513: also slope can have a big hydrological influence, potentially also soil freezing.*

Added as suggested:

**"Still, other factors, such as hillslope effects, soil characteristics, and soil frost, also affect how heterogeneous snow melt runoff in montane forests translates into a hydrograph response (Pomeroy et al., 2012; Redding and Devito, 2011)."**

*L525: 15 years is not that long ago, I'd say computational snow hydrology in forests dates well beyond that in for example snow interception work.*

We reformulated this sentence addressing corresponding comments of both reviewers 1 and 2:

**"Throughout the past decades, modeling studies have played an important role in shaping today's understanding of both specific forest snow processes (e.g., Pomeroy et al., 1998) and how process interactions in montane forests modulate the subcanopy snow cover (e.g., Pomeroy et al., 2012; Strasser et al., 2011)."**

*L540-546: How about edge effects in snow being accumulated at the transition between open areas and forest stands due to changing wind fields. Are they too small scale of a process for your analysis?*

Edge effects are accounted for in the model within a buffer of 25m outside the forest. Note that for our study region, such edge effects are mainly driven by thermal radiation. Tests have shown that upscaled meter-scale resolutions that explicitly resolve these edge effects match results from coarser resolution simulation like the ones used here, c.f. Mazzotti et al. (2021).

*L564: You don't really analyze runoff generation, so this should be written in a more speculative tone. In particular as its part of conclusions chapter.*

Yes, thanks for the comment, we totally agree. We revised the statement to now read:

**"... with potential consequences for runoff generation from snowmelt at catchment scales (Figs. 8 and 10)."**

**References**

Hedstrom, N. R. and Pomeroy, J. W.: Measurements and modelling of snow interception in the boreal forest, Hydrol. Process., 12, 1611–1625, https://doi.org/10.1002/(SICI)1099-1085(199808/09)12:10/11%253C1611::AID-HYP684%253E3.0.CO;2-4, 1998.

Mazzotti, G., Essery, R., Webster, C., Malle, J., and Jonas, T.: Process-Level Evaluation of a Hyper-Resolution Forest Snow Model Using Distributed Multisensor Observations, Water Resour. Res., 56, e2020WR027572, https://doi.org/10.1029/2020WR027572, 2020a.

Mazzotti, G., Essery, R., Moeser, C. D., and Jonas, T.: Resolving Small-Scale Forest Snow Patterns Using an Energy Balance Snow Model With a One-Layer Canopy, Water Resour. Res., 56, e2019WR026129, https://doi.org/10.1029/2019WR026129, 2020b.

Mazzotti, G., Webster, C., Essery, R., and Jonas, T.: Increasing the Physical Representation of Forest-Snow Processes in Coarse-Resolution Models: Lessons Learned From Upscaling Hyper-Resolution Simulations, Water Resour. Res., 57, e2020WR029064, https://doi.org/10.1029/2020WR029064, 2021.

Mazzotti, G., Webster, C., Quéno, L., Cluzet, B., and Jonas, T.: Canopy structure, topography, and weather are equally important drivers of small-scale snow cover dynamics in sub-alpine forests, Hydrol. Earth Syst. Sci., 27, 2099–2121, https://doi.org/10.5194/hess-27-2099-2023, 2023.

Mott, R., Winstral, A., Cluzet, B., Helbig, N., Magnusson, J., Mazzotti, G., Quéno, L., Schirmer, M., Webster, C., and Jonas, T.: Operational snow-hydrological modeling for Switzerland, Front. Earth Sci., 11, https://doi.org/10.3389/feart.2023.1228158, 2023.

Pomeroy, J., Fang, X., and Ellis, C.: Sensitivity of snowmelt hydrology in Marmot Creek, Alberta, to forest cover disturbance, Hydrol. Process., 26, 1891–1904, https://doi.org/10.1002/hyp.9248, 2012.

Pomeroy, J. W., Parviainen, J., Hedstrom, N., and Gray, D. M.: Coupled modelling of forest snow interception and sublimation, Hydrol. Process., 12, 2317–2337, https://doi.org/10.1002/(SICI)1099-1085(199812)12:15%253C2317::AID-HYP799%253E3.0.CO;2-X, 1998.

Redding, T. and Devito, K.: Aspect and soil textural controls on snowmelt runoff on forested Boreal Plain hillslopes, Hydrol. Res., 42, 250–267, https://doi.org/10.2166/nh.2011.162, 2011.

Strasser, U., Warscher, M., and Liston, G. E.: Modeling Snow–Canopy Processes on an Idealized Mountain, J. Hydrometeorol., 12, 663–677, https://doi.org/10.1175/2011JHM1344.1, 2011.